# Impact of reservoir evaporation on future water availability in North-Eastern Brazil: A multi-scenario assessment

Gláuber P. Rodrigues[1,2], Arlena Brosinsky[2,3], Ítalo S. Rodrigues[4], George L. Mamede[5], José C. de Araújo[1]

[1] Department of Agricultural Engineering, Federal University of Ceará, 60451-070 Fortaleza, Brazil

5 [2] Institute for Environmental Sciences and Geography, University of Potsdam, Potsdam, Germany

[3] Remote Sensing and Geoinformatics Section, German Research Centre for Geosciences (GFZ), Potsdam, Germany

[4] Department of Geography, University of Lethbridge, Lethbridge, Alberta, Canada

[5] Institute for Engineering and Sustainable Development, University of International Afro-Brazilian Integration, 62.790-000 Redenção, Brazil.

10 *Correspondence to*: Gláuber P. Rodrigues (pontesglauber@gmail.com)

**Abstract.** The potential effects of climatic changes on water resources are crucial to be assessed, particularly in dry regions such as Northeast Brazil (1 million km²), where water supply is highly reliant on open-water reservoirs. This study analyses the impact of evaporation (by the Penman method) on water availability for four scenarios based on two regional climatic models (Eta-CanESM2 and Eta-MIROC5) under the Representative Concentration Pathways 4.5 and 8.5. We compared the 15 water availability in the period of 2071-2100 with that of the historical period (1961-2005). The scenarios derived from the Eta-CanESM2 model indicate an increase in dry season evaporative rate (2% and 6%, respectively) by the end of the century. Unlike the above scenarios, the ones derived from the Eta-MIROC5 model both show a decrease in dry season evaporative rate of -2%. Consequently, for a 90% reliability level, the expected reservoir capacity to supply water with high reliability is reduced in 80%. It is reasonable to state that both patterns of future evaporation in the reservoirs may prove to be plausible. 20 Because model-based projections of climate impact on water resources can be quite divergent, it is necessary to develop adaptations that do not need quantitative projections of changes in hydrological variables, but rather ranges of projected values. Our analysis shows how open water reservoirs might be impacted by climate change in dry regions. These findings complement a body of knowledge on estimation of water availability in a changing climate and provide new data and insights on water management in reservoir-dependent drylands.

# 1 Introduction

The increasing atmospheric concentration of greenhouse gases is changing the Earth's climate more rapidly than ever before (Konapala et al., 2020). Expected changes in climate variables, such as rainfall and temperature, may result in alterations in the hydrological cycle and, thus, the necessity of adaptations of current reservoir management strategies (Minville et al. 2010). As climate changes, it is imperative to identify its impact on water supply, especially with respect to open-water reservoirs 30 located in drylands.

As stated by Adrian et al. (2009), lakes, reservoirs and other inland open-water surfaces are likely to serve as good sentinels for current climate change because (1) aquatic ecosystems are well defined and are studied in a continuous way; (2) they respond directly to climate change and also incorporate the effects of climate-driven changes occurring within the catchment; (3) they integrate responses over time, which can filter out random noise; and (4) they are distributed worldwide and, as such, can act as sentinels in many different geographic locations and climatic regions, capturing different aspects of climate change (e.g., rising temperature, glacier retreats, permafrost melting). Indeed, several investigations worldwide have highlighted the impact of climate change on water resources. Reservoirs are essential infrastructures for the economic and social development of the region, and their evaporation losses are significant for the water system and can severely impact water availability and allocation (Malveira et al., 2012; Mamede et al., 2012; Peter et al., 2014).

In hydrology and in studies relating to water availability, free-surface evaporation plays an important role. However, despite numerous approaches developed over the last 200 years to estimate evaporation (McMahon et al., 2016), there are still uncertainties within concerning evaporation assessment, and the main reasons for this are: the high cost of maintaining the equipment (including personnel training); the numerous parameters needed to apply the equations; and the use of databases located far from the respective water bodies (for instance, class-A pans or meteorological stations). Additionally, remote sensing tools can assist in monitoring evaporation (Cui et al., 2019) and water volume loss in general (Zhang et al., 2021). As stated by (Su, 2002), conventional techniques that employ point measurements to estimate the components of energy balance are representative only of local scales and cannot be extended to large areas because of the heterogeneity of land surfaces and the dynamic nature of heat transfer processes. Remote sensing is probably the only technique which can provide representative measurements of several relevant physical parameters at scales from a point to a continent. Techniques using remote sensing information to estimate atmospheric turbulent fluxes are therefore essential when dealing with processes that cannot be represented by point measurements only.

Climate change plays a critical role in the planning of water resources: the water cycle is expected to be accelerated because of temperature increase; in warm climates, climate change is expected to worsen water shortage episodes. Investigations show the sensitivity of inland water bodies as physical, chemical, and biological water properties respond rapidly to climate-related changes (Adrian et al., 2009; Rosenzweig et al. 2007). Climate change is studied using global circulation models, usually with a coarse spatial resolution of hundreds of kilometres, disregarding regional factors in the process of modelling (Chou et al., 2014; Navarro-Racines et al., 2020). Hence, the importance of regional climate models (RCMs), which have a finer resolution (usually tens of kilometres) and consider local factors such as topography, land cover, and land use. One of the issues addressed refers to the resilience of metropolitan areas to climate change, where major problems may be related to rapid changes in water supply, among others (Lyra et al., 2018).

The Metropolitan Region of Fortaleza (MRF) has a population of over 4 million inhabitants (IBGE, 2022), one of the most densely populated areas in Brazil. Its water supply is highly dependent on an extensive network of open-water reservoirs, almost entirely located in the semiarid region (Mamede et al., 2018; Peter et al., 2014). Despite the well-known sensitivity of dry regions to changes in climate, little is yet known about the impacts on water resources in this region and how water

availability may be affected by climate change. With this investigation, we intend to increase the amount of information currently existing on the potential impacts of climate change on water resources in Northeast Brazil. After several studies in this region have focused on temperature variation (Marengo et al., 2009) and precipitation (Almagro et al., 2020), our analysis will contribute to the growing body of knowledge on surface water reservoirs and how they respond to climate change.

More specifically, the objective of this research was to analyse the impact of climate change on reservoir evaporation and
consequently on water availability in the MRF, a densely populated urban area located in North-eastern Brazil. To achieve this goal, we (i) assessed historical and present evaporation by means of a weather station and remote sensing calculations, (ii) compared them to historical and future simulations based on four climate change scenarios, and (iii) investigated the impact of evaporation change on water availability using stochastic modelling.

**2 Study area**

The state of Ceará is situated in the north-eastern region of Brazil (Figure 1), mostly semiarid (BSh, according to Köppen classification), a dryland where various studies on the impacts of climate change have shown susceptibility to severe climate extremes, more prominently droughts (Alvalá et al., 2019; Marengo et al., 2017, 2022; Vieira et al., 2020). The average rainfall in the region ranges from 500 to 800 mm per year (of which 80% occurs between January and April) and the average annual
temperature is 26 °C (INMET, 2019; Rodrigues et al., 2021a). This region has high water vulnerability resulting from irregular rainfall associated with ephemeral rivers, high potential evaporation rates exceeding 2000 mm per year, and shallow soils over crystalline basement (de Araújo and Piedra, 2009; Medeiros and de Araújo, 2014) . This soil-feature allows limited storage of water (Alvalá et al., 2019), which when present is often salty because of the prevalence of fissural aquifers in crystalline bedrock.

The reservoirs studied in this research are located in a tropical-coastal area whose climate is slightly different from the predominant semi-arid condition described above (rainfall of 1,600 mm year-1 and relative humidity of 78%) – however, they are supplied by a water system from the drylands.

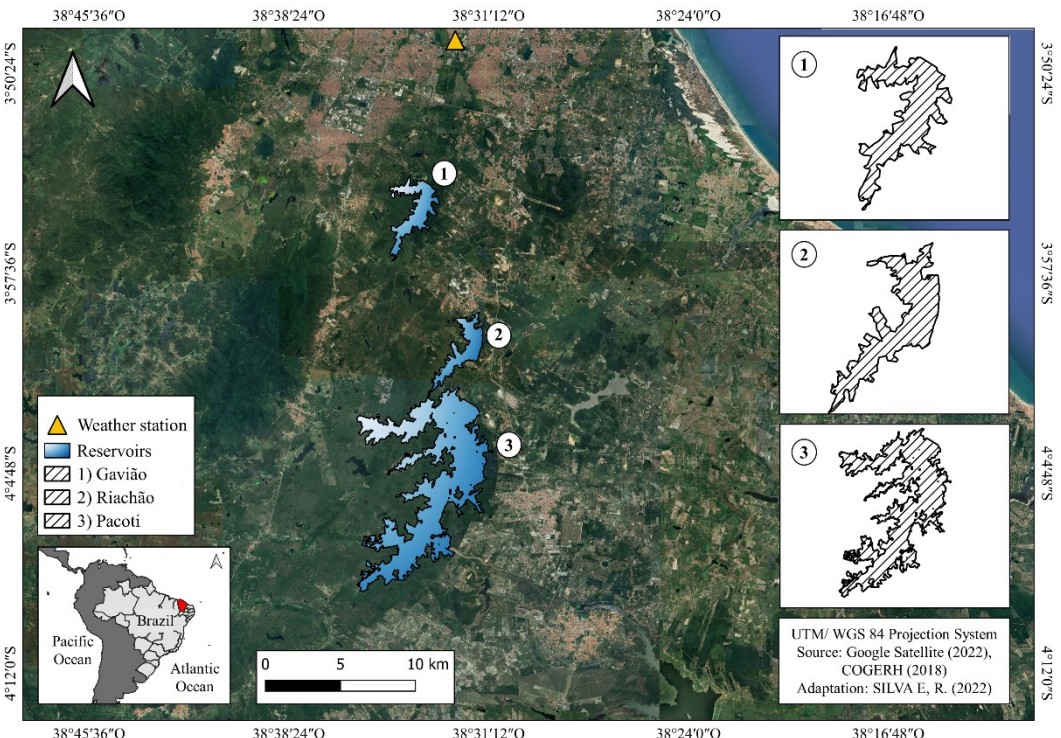

**Figure 1: Location of the state of Ceará (bottom left) and the major reservoirs that supply the Metropolitan Region of Fortaleza, the hydrographic network, and the INMET (Brazilian National Institute of Meteorology) weather station. Daily meteorological observed variables from the station were used spanning 1961 to 2005.**

The reservoirs under analysis (Pacoti, Riachão and Gavião, see Table 1 and Figure 1) are responsible for supplying water to the 4.2 million inhabitants of the MRF region and comprise the downstream sector of a network of reservoirs located in the sub-humid area. Unlike the reservoirs that supply them, the water level of the three reservoirs does not decrease substantially during the dry season, due to an interbasin transfer that provides water.

**Table 1:** Technical characteristics of Pacoti, Riachão, and Gavião reservoirs.

|  | Pacoti | Riachão | Gavião |
|---|---|---|---|
| Storage capacity (hm³) | 380.0 | 47.9 | 33.3 |
| Catchment area (km²) | 1,110 | 34 | 97 |
| Maximum water surface area (km²) | 37.0 | 5.7 | 6.2 |

Source: Ceará State Secretariat for Water Resources, SRH (2015).

## 3 Methodology

We used the regional Eta model (Mesinger et al. 2012) nested to two Global Circulation Models (CanESM-2 and MIROC5) at Representative Concentration Pathways 4.5 and 8.5. The investigation was performed in three phases: the first consisted of the simulation of regional evaporation pattern with bias correction; the second consisted of the combined application of remote sensing and on-site measurement to assess open-water evaporation; and on the third, we analysed the impact of evaporation changes on water availability for four scenarios. For ease and convenience, the four scenarios are hereafter referred to as C4 (model Eta-CanESM2, Pathway 4.5), C8 (model Eta-CanESM2, Pathway 8.5), M4 (model Eta-MIROC5, Pathway 4.5), and M8 (model Eta-MIROC5, Pathway 8.5). The methodological framework is presented in Figure 2. All climate projections set time slices as follows: historical (1961-2005), near term (2006-2040), midterm (2041-2070), and long term (2071-2099).

It is outside the scope of this work to study the effects on the general hydrology of the MRF, but rather focus on a single hydrological process, which is a major cause of water losses in the whole region.

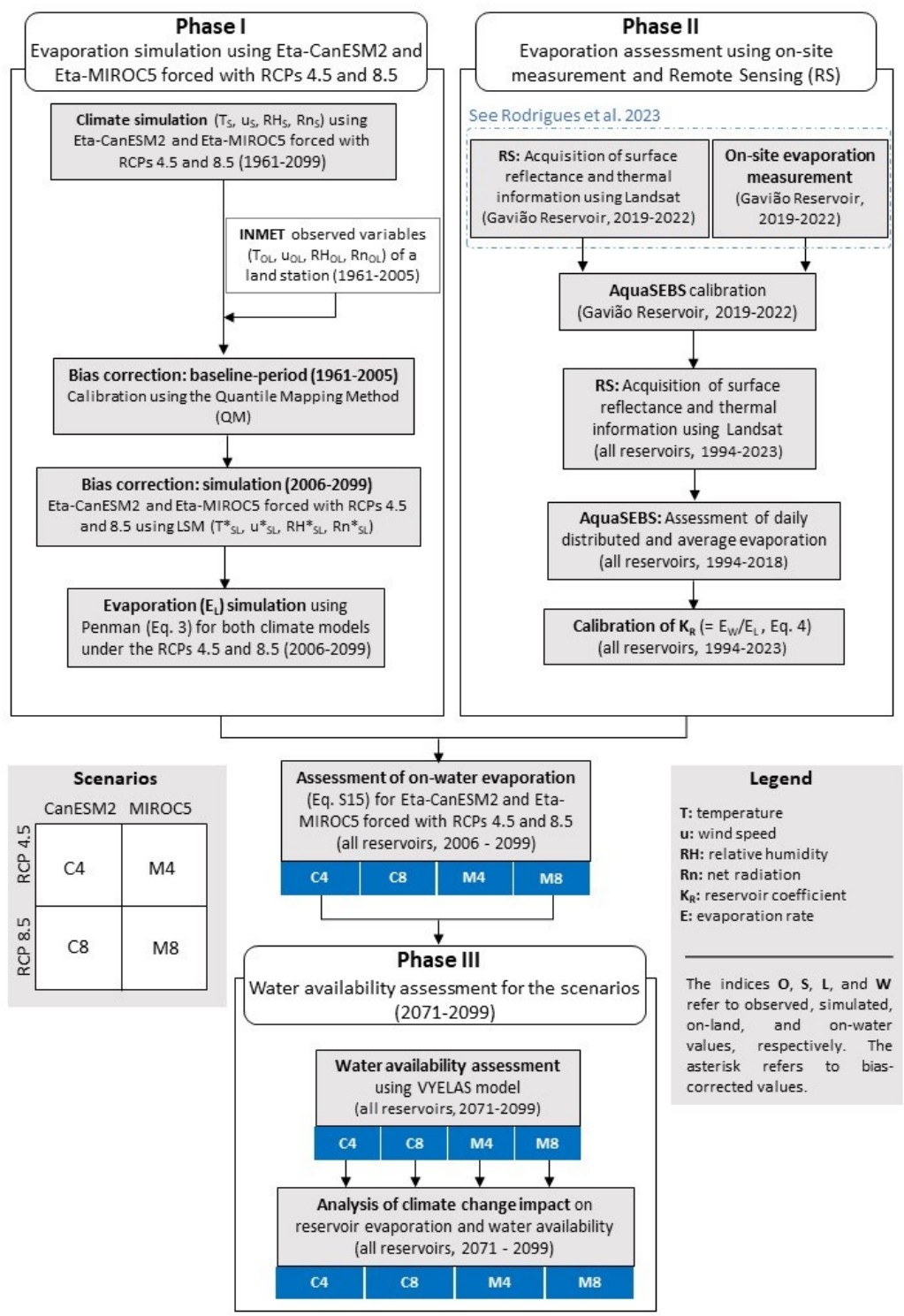

**Figure 2: Methodological flowchart for the simulation of evaporation with climate models and assessment of water availability.**

## 3.1 Phase I: Evaporation simulation forced with RCPs 4.5 and 8.5

The database used in this research is provided by Brazilian National Institute for Space Research (INPE) on the PROJETA platform (see https://projeta.cptec.inpe.br). That is the official source of downscaled climate data for the Brazilian territory with the Eta Regional Model (Mesinger et al., 2014). The latest version of the platform provides parameters downscaled from three global circulation models (CanESM-2, MIROC5 and HadGEM2-ES) forced with the Eta Model with a spatial resolution of 20 km for the study area. The Representative Concentration Pathways available on the platform are 4.5 and 8.5. According to Chou et al. (2014), the Brazilian model has been used operationally at INPE since 1997 for weather forecasts, and since 2002 for seasonal climate forecasts.

Since the Fifth Assessment Report (AR5) from the Intergovernmental Panel on Climate Change (IPCC 2014), the greenhouse gas concentration scenarios are based on the Representative Concentration Pathways (RCP), which are expressed in terms of radiative forcing toward the end of the twenty-first century. In this study, the downscaled products were simulated based on RCP4.5 and RCP8.5. The first provides a climate forcing that reaches 4.5 W m$^{-2}$ by 2050, with further stabilization; the latter provides continued growth of the radiative forcing to reach 8.5 W m$^{-2}$ by 2100 (Bjørnæs 2013).

Outputs from regional climate models are often prone to systematic errors (biases), therefore, bias correction is recommended (Graham et al., 2007) for simulation reliability purposes (Teutschbein and Seibert, 2012). To correct bias, we applied the Quantile-mapping method, which is widely used in climate and hydrological modelling (Thrasher et al., 2012; Teutschbein and Seibert, 2012) to improve the accuracy and reliability of model simulations, particularly in regions or variables where models exhibit systematic biases. It is a flexible and computationally efficient method that can be applied to a wide range of climate-related variables and datasets.

## 3.2 Phase II: Evaporation assessment using on-site measurement and Remote Sensing

### 3.2.1 On-land ($E_L$) evaporation assessment

Daily meteorological observed variables from a station of the Brazilian National Institute of Meteorology (INMET) were used for bias correction of the climate model for the period from 1961 to 2005 (station code: 82397 - Fortaleza). The station was chosen due to two aspects: it is the nearest weather station to the reservoirs (15 km away), it boasts a relatively extensive historical dataset dating back to 1961, exhibiting fewer flaws compared to neighbouring stations. To assess on-land evaporation, we used the Penman (1948) Equation 1:

$$E_L = \frac{\Delta}{\Delta + \gamma} \cdot \frac{R_n}{\lambda v} + \frac{\gamma}{\Delta + \gamma} \cdot f(u) \cdot (e_s - e_a) \qquad (1)$$

In Equation 3, $E_L$ is the open-water evaporation rate (mm d$^{-1}$); $R_n$ is the net radiation at the water surface (MJ m$^{-2}$ d$^{-1}$); $\Delta$ is the slope of the saturation vapour pressure curve (kPa ºC$^{-1}$) at air temperature; ($e_s$ - $e_a$) is the difference between saturation and partial water vapour pressure (kPa), $\gamma$ is the psychrometric coefficient; $\rho$ is the density of water (1000 kg m$^{-3}$); $\lambda v$ is the latent heat of vaporization. The procedure to obtain $\gamma$ and $\lambda$ is described in Annex 2 of the Food and Agriculture Organization of the United Nations (FAO) 56 protocol (Allen et al. 1998). In Equation 1, f(u) is a function used to account for the advective drying effects of wind (mm d$^{-1}$kPa$^{-1}$). In Equation 2, u is the wind speed at 2 m height (m s$^{-1}$). We have adopted the Penman (1956) form of the wind function:

$$f(u) = 1.313 + 1.381 \cdot u \tag{2}$$

A trend analysis was performed using the Mann-Kendall method (Kendall, 1975; Mann, 1945) applied to both the calculated and the simulated Penman-evaporation. The null hypothesis is that there is no trend in the series. The three hypotheses evaluated are: i) no trend, ii) positive trend, iii) negative trend. A significance level of p = 0.05 was adopted. The magnitude of the changes was evaluated by the nonparametric Sen's slope and Kendall's tau ($\tau$) coefficient, which describes the relationship between variables.

### 3.2.2 On-water ($E_W$) evaporation assessment

The on-water evaporation rate was estimated with the remote sensing algorithm AquaSEBS (Surface Energy Balance of Fresh and Saline Waters, see Abdelrady *et al.*, 2016). The algorithm is a modification from Su (2002) and was developed to estimate the heat fluxes by integrating satellite data and hydro-meteorological field data. It has been validated on the study region (Rodrigues *et al.*, 2021a, 2021b), and requires three sets of data as input: (i) remote-sensing data, including emissivity, surface albedo and surface temperature; (ii) meteorological data; and (iii) radiative forcing parameters (Abdelrady et al., 2016), such as downward shortwave and long-wave radiations. For the temporal estimation of evaporation, were used: bands 1 to 5 (reflectance) and band 6 (thermal) from Landsat 5 (Thematic Mapper - TM) and bands 1 to 7 (reflectance) and band 10 (thermal) from Landsat 8 (Optical Land Imager - OLI). Due to the technical characteristics of the sensors (radiometric, spectral, and thermal band spatial resolutions), methodological adjustments were necessary to acquire some parameters in the model application (see Supplementary Material 1).

AquaSEBS uses energy balance to calculate instantaneous latent heat flux of evaporation (Equation 3), thus, evaporation is calculated for each pixel of the image. The energy balance of the water surface can be expressed as it follows:

$$E_w = \frac{R_n - G_w - H}{\lambda v \cdot \rho} \tag{3}$$

In Equation 3, $R_n$ is net radiation at the water surface, $G_w$ is the water (or ground for land surfaces) heat flux, and H is sensible heat to the air. All terms are expressed in W m$^{-2}$. Detailed information about the algorithm and the meteorological input is in the Supplementary Material 1 of this manuscript. To assess the on-water evaporation rate ($E_w$) in the scenarios, we used Equation 4, in which $E_L$ is the on-land evaporation rate and $K_R$ is a coefficient. $E_L$ refers to a value provided by the models

Eta-CanESM2 and Eta-MIROC5, whose bias was corrected using data from an on-land meteorological station. The coefficient $K_R$ was calibrated using AquaSEBS to assess $E_w$ and Penman Equation to estimate $E_L$ based on data from the INMET station.

$$E_w = K_R \cdot E_L \qquad (4)$$

For calibration purposes, AquaSEBS used images spanning from 1994 to 2023 containing the three studied reservoirs. A total of 35 scenes were used, eighteen from Landsat 5 and seventeen from Landsat 8 (Figure 3), all acquired from the United States Geological Survey portal (https://earthexplorer.usgs.gov/, last accessed 10 March 2024). We used images exclusively from the dry season (June to December) due to the following reasons: (i) cloud-free images are easier to obtain in these months; (ii) the water-availability model (Phase III) only considers evaporation of the dry period; and (iii) this is the period when
evaporation is more intense and, thus, more relevant to water management purposes.

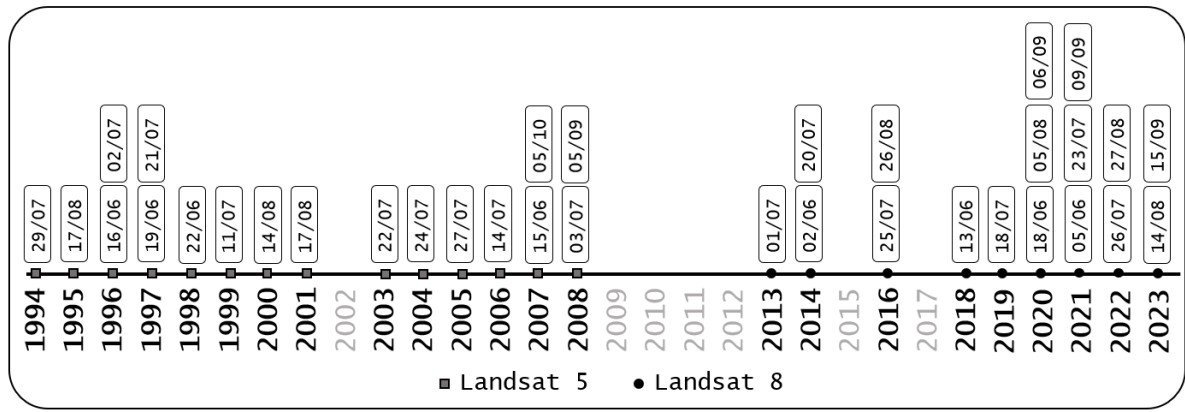

**Figure 3: Temporal distribution of Landsat images used to assess on-water evaporation with the AquaSEBS algorithm and, thus, to calibrate the KR coefficient. Years shown in grey did not have cloud-free images.**

**3.3 Phase III: Water availability assessment for the scenarios in the long-term future (2071-2099)**

     The VYELAS model (Volume-Yield-Elasticity, as in de Araújo *et al.*, 2006) was applied to simulate water availability. River inflow to the reservoir is generated by a stochastic procedure, using the inverse of the two-parameter gamma
probability density function (see McMahon and Mein, 1986; and Campos, 1996). The parameters of the distribution were derived from the average and standard deviation of historical annual inflow to the reservoir. A 10,000-year synthetic series was generated for each of the three reservoirs, which reproduced the historical average and the coefficient of variation of annual inflow as given in Table 2. VYELAS simulates the reservoir water balance for a large synthetic series, corresponding to the number of annual water-balance simulations (Equation 5). Each set of simulations is associated with a withdrawal
discharge ($Q_w$). The model calculates the annual reliability level (G), which corresponds to the annual probability of providing

the target withdrawal discharge and is given by $G = 1 - N_S/N$, where $N_S$ is the number of successful years and N the total number of years in the simulation (Campos, 2010; McMahon and Mein, 1986). In this context, a successful year is one in which the target water withdrawal can be integrally met without leading the reservoir reserve below the minimum operational volume (de Araújo et al., 2018). Campos (2010) states that the water balance of reservoirs in semiarid environments is approximately:

$$\frac{\Delta V}{\Delta t} \approx Q_{in} - Q_{E,\,dry} - Q_s - Q_w \tag{5}$$

In Equation 5, V is the water storage volume in the reservoir, t represents time, $Q_{in}$ the inflow from the river network into the reservoir, $Q_{E,dry}$ the water loss due to evaporation in the dry season, $Q_S$ the reservoir outflow over the spillway, and $Q_w$ the water withdrawal from the reservoir (all variables in hm³ year⁻¹). The model operates under the assumption that the combined water input from rainfall directly onto the reservoir surface and groundwater discharge into the reservoir is offset by wet season evaporation and outflow resulting from seepage, thus rendering rainfall as negligible in its overall impact. VYELAS demands data of seasonal water inflow (average and standard deviation), precipitation, evaporation, storage capacity (SC), alert volume[1], and the morphological parameter $\alpha$ ($SC = \alpha \cdot y^3$, where y is the water maximum depth) (Campos, 2010).

The evaporation rates simulated for the long-term future (last 30 years of the century) of the two models were adjusted by the $K_R$ coefficient (see Equation 4) and used as input in the VYELAS model. The input data required to run the model are listed in Table 2.

**Table 2.** Input data for the VYELAS model for the three reservoirs.

| | Gavião | Riachão | Pacoti |
|---|---|---|---|
| Average inflow (hm³ yr⁻¹) [a] | 32.6 | 7.8 | 254.5 |
| Coefficient of variation annual inflow [b] | 0.8 | 0.8 | 0.8 |
| Reservoir-shape coefficient [c] | 17927 | 5007 | 31174 |
| Evaporation in the dry season (m yr⁻¹) [d] | 1.3 | 1.3 | 1.3 |
| Maximum storage capacity (hm³) [a] | 33.3 | 47.9 | 380.0 |
| Minimum operational volume (hm³)[*] | 5.0 | 7.2 | 57.0 |
| Initial volume in the first simulation year (hm³)[**] | 16.3 | 3.9 | 127.2 |

Source of data: [a]COGERH (2020); [b]Macêdo (1981); [c]Feitosa et al. (2021); [d]Calculated in this study.
[*]The minimum operational volume was assumed as 15% of the reservoir storage capacity (de Araújo *et al.*, 2018).
[**]Initial volume is the smallest value between half of maximum storage capacity and half of the annual average inflow (Campos, 2010)

---

[1] The "alert volume" is the critical water level that prompts water management actions. Typically, a fraction of the reservoir's total volume, it signals when water availability becomes concerning. At this point, measures like usage restrictions, conservation plans, or emergency actions are activated to ensure a continuous water supply for users and ecosystem preservation. In Brazil's semi-arid region, this volume is usually set at 5% of the reservoir's total capacity.

To assess how water availability varies with evaporation rate changes, we use the concept of elasticity ($\varepsilon$, as in de Araújo et al., 2006; 2018), represented by Equation 6, in which $Q_{90}$ is the water availability with 90% annual reliability and E is the evaporation rate. Therefore, the higher the elasticity, the more significant the impact of evaporation on water availability. The asterisk refers to the reference values.

$$\varepsilon(Q_{90}:E) = \frac{\Delta Q_{90}/Q_{90}^*}{\Delta E/E^*} \tag{6}$$

The VYELAS model has been used in hydrological studies assessing the effects of water quality on evaporation (Mesquita et al., 2020), reservoir operating rules (de Araújo et al., 2018), reservoir water balance (Feitosa *et al.*, 2021) and reservoir silting (de Araújo *et al.*, 2006, López-Gil *et al.*, 2020).

## 4 Results

### 4.1 Spatialised evaporation rate in the reservoirs with AquaSEBS ($E_w$)

Figure 4 shows the spatialised evaporation in the Gavião, Pacoti and Riachão reservoirs. Within-reservoir variability cannot be seen very clearly in the chosen range, which was selected for a better comparison between acquisitions over the years. It is noticeable that over the last decades of monitoring, lower evaporation rates (greenish tones) occur more frequently. The highest evaporation rates are found from 1994 to 2000, which would suggest a declining evaporative pattern in the reservoirs over the last decades. However, we cannot affirm that this apparent decrease represents the reality of the reservoirs, since there are on average only two cloud-free Landsat images per year and there is a degree of uncertainty in the extrapolation of the satellite passage time (instantaneous evaporation) for 24-hour evaporation. From 2000 onwards, one can notice scenes where the evaporation rate is low (about 3 mm day$^{-1}$), as in the scene obtained in the year 2004. This comes as a result of atypically rainy years in the Ceará state (Medeiros and de Araújo, 2014; Medeiros and Sivapalan, 2020), as the abovementioned year, when the historical average rainfall (800 mm yr$^{-1}$) was exceeded by over 40% (FUNCEME, 2023). While the images were captured during the dry season when cloud cover was minimal, it is important to acknowledge that variations in surface temperature and relative humidity could have influenced the algorithm's results.

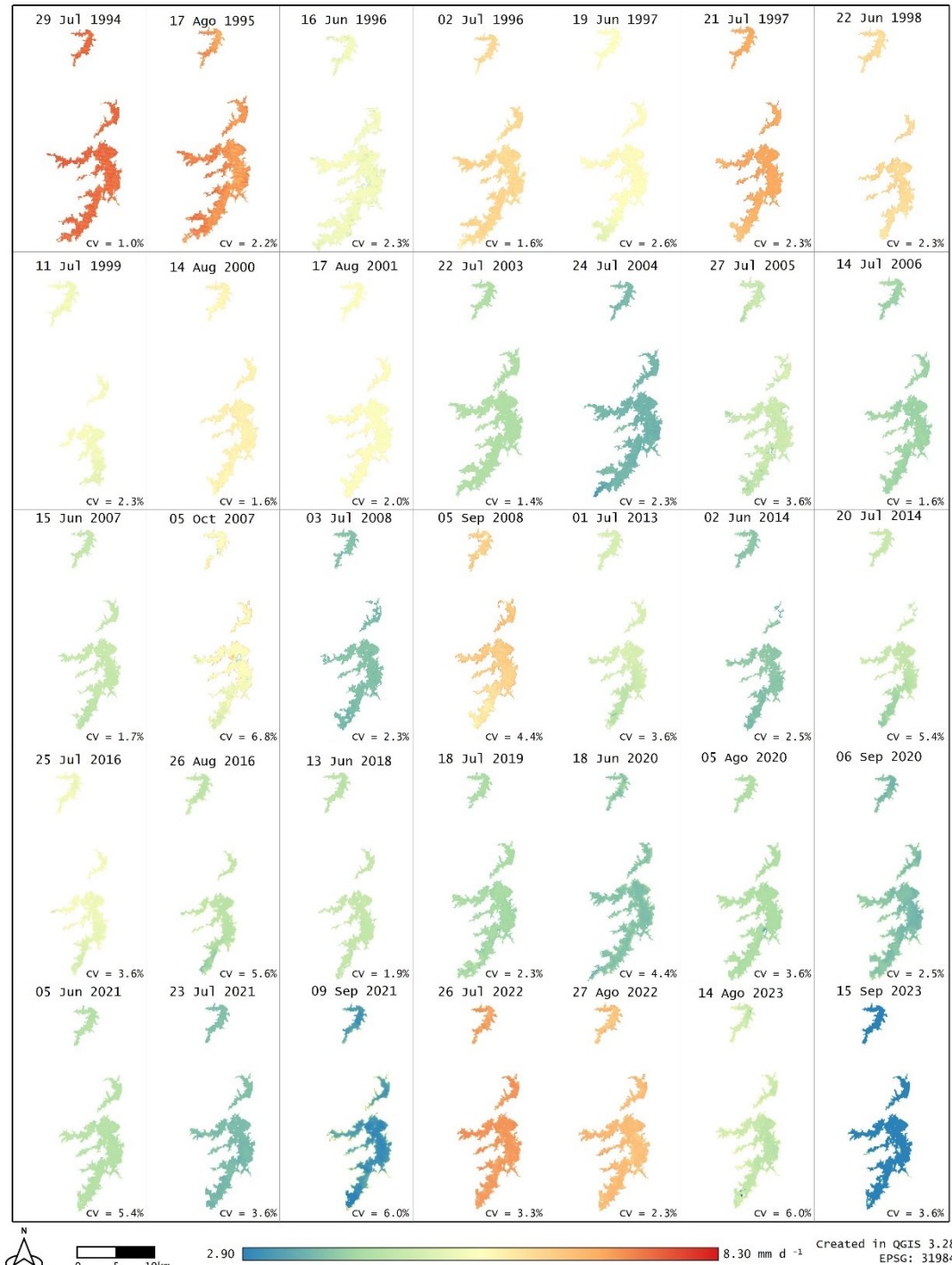

**Figure 4: Evaporation rates in the Fortaleza Metropolitan Region reservoirs (Gavião, Riachão and Pacoti) obtained with AquaSEBS algorithm for 35 individual days in the period 1994-2023.**

Different reservoirs have different evaporation rates due to several drivers, such as depth, surface area and water transparency. The first two factors influence the change in water heat storage and the phase lag between energy and evaporation rate; the latter is directly related to the albedo, which depends on the water quality and changes the reflection properties of the surface (Mesquita et al., 2020; McMahon et al., 2016). However, after evaluating the reservoir pixels with AquaSEBS, we observed that the daily evaporation rates and the coefficient of variation (see Figure 4) did not differ substantially, regardless of whether the reservoir evaporation were assumed separately or as a single raster. Thus, we assumed that the reservoirs are subject to the same evaporation rate. Regarding the spatial variability of evaporation within the water bodies, it is necessary to highlight the uncertainty in the result related to the only-water pixels. This is caused by the presence of aquatic plants or exposed soil on some of the banks of the reservoirs, which are captured by the Landsat pixels (spatial resolution = 30 m), thus, hindering the accuracy of the assessment.

## 4.2 Comparison between on-land evaporation (EL) and remote-sensed evaporation assessment

Table 3 shows the comparison between the on-land ($E_L$) and on-water ($E_w$) evaporation rates. In general, the average daily evaporation rates differ by 27%, in which the on-land evaporation rate is constantly higher than the on-water evaporation rate: $E_w$ averages 5.10 mm d$^{-1}$ against an average $E_L$ of 7.04 mm d$^{-1}$ (Figure 5); while the correction value $K_R$ averages 0.73. We also examined the data in order to detect a possible correlation of $K_R$ values with the period of the year when evaporation rates were estimated (for example, higher ratios at the end of the dry season). However, no correlation was found between the coefficient and the period of assessment. Most of the highest $K_R$ values (above 0.85) were registered in the first years of monitoring; six of the seven ratios at this threshold are from before 1999. Future investigations may investigate the correlation of this with factors not addressed in this paper, such as features on the satellite sensor, for instance.

**Table 3.** Evaporation in Gavião, Riachão and Pacoti reservoirs (1994-2023): comparative results of on-water ($E_w$) and on-land ($E_L$) daily rates. The lower part of the table shows statistical parameters.

| n | Date | Pixel count[*] | $E_L$ (mm d$^{-1}$) | $E_w$ (mm d$^{-1}$)[**] | $K_R$ |
|---|---|---|---|---|---|
| 1 | 29/07/1994 | 27350 | 6.89 | 6.65 | 0.96 |
| 2 | 17/08/1995 | 48980 | 8.08 | 7.19 | 0.89 |
| 3 | 16/06/1996 | 57108 | 7.23 | 5.40 | 0.75 |
| 4 | 02/07/1996 | 47735 | 6.92 | 6.24 | 0.90 |
| 5 | 19/06/1997 | 38887 | 6.98 | 5.61 | 0.80 |
| 6 | 21/07/1997 | 36370 | 7.69 | 6.97 | 0.91 |
| 7 | 22/06/1998 | 23277 | 7.25 | 6.23 | 0.86 |
| 8 | 11/07/1999 | 17344 | 7.39 | 5.43 | 0.73 |
| 9 | 14/08/2000 | 34668 | 8.31 | 5.82 | 0.70 |

| | | | | | |
|---|---|---|---|---|---|
| 10 | 17/08/2001 | 32373 | 8.24 | 5.61 | 0.68 |
| 11 | 22/07/2003 | 55487 | 7.31 | 4.37 | 0.60 |
| 12 | 24/07/2004 | 53372 | 7.18 | 3.70 | 0.52 |
| 13 | 27/07/2005 | 41088 | 7.54 | 4.74 | 0.63 |
| 14 | 14/07/2006 | 39100 | 7.16 | 4.15 | 0.58 |
| 15 | 15/06/2007 | 38071 | 7.47 | 4.70 | 0.63 |
| 16 | 05/10/2007 | 34419 | 8.48 | 5.60 | 0.66 |
| 17 | 03/07/2008 | 38744 | 6.93 | 3.84 | 0.55 |
| 18 | 05/09/2008 | 41943 | 8.55 | 6.38 | 0.75 |
| 19 | *01/07/2013* | 35519 | 6.19 | 4.88 | 0.79 |
| 20 | *02/06/2014* | 31122 | 6.09 | 3.94 | 0.65 |
| 21 | *20/07/2014* | 30688 | 6.20 | 4.60 | 0.74 |
| 22 | *25/07/2016* | 26976 | 6.20 | 5.38 | 0.87 |
| 23 | *26/08/2016* | 27823 | 6.80 | 4.62 | 0.68 |
| 24 | *13/06/2018* | 28585 | 6.21 | 4.69 | 0.75 |
| 25 | *18/07/2019* | 32195 | 6.09 | 3.93 | 0.65 |
| 26 | *18/06/2020* | 43239 | 5.50 | 3.60 | 0.65 |
| 27 | *05/08/2020* | 30653 | 6.56 | 4.07 | 0.62 |
| 28 | *06/09/2020* | 34946 | 6.67 | 3.58 | 0.54 |
| 29 | *05/06/2021* | 46733 | 4.88 | 4.17 | 0.85 |
| 30 | *23/07/2021* | 37400 | 7.43 | 3.47 | 0.47 |
| 31 | *09/09/2021* | 48203 | 8.01 | 4.26 | 0.53 |
| 32 | *26/07/2022* | 48284 | 5.94 | 7.12 | 1.20 |
| 33 | *27/08/2022* | 31249 | 7.05 | 9.12 | 1.29 |
| 34 | *14/08/2023* | 27358 | 7.83 | 5.59 | 0.71 |
| 35 | *15/09/2023* | 31505 | 7.10 | 3.00 | 0.42 |
| | Average | 36405 | 7.04 | 5.10 | 0.73 |
| | Median | 35519 | 7.10 | 4.74 | 0.70 |
| | Min | 2735 | 4.88 | 3.00 | 0.42 |
| | Max | 57108 | 8.55 | 9.12 | 1.29 |
| | Std | 10899 | 0.85 | 1.32 | 0.18 |
| | CV | 0.30 | 0.12 | 0.26 | 0.25 |

\* Number of reservoir pixels in the raster
\*\* Each value refers to the average of the pixels in the three reservoirs

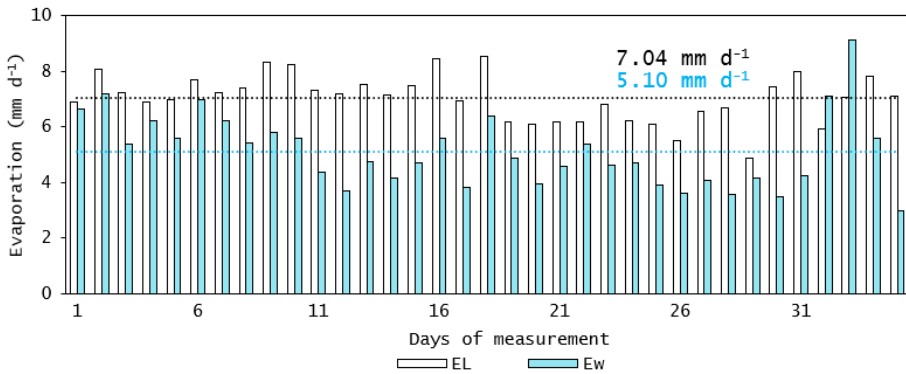

**Figure 5: Daily evaporation estimated with the AquaSEBS algorithm (Ew) and calculated based on variables obtained from the INMET station (EL). The analysed period is from 1994 – 2023, 35 days of measurement.**

## 4.3 Evaporation simulation under four climate change scenarios

Figure 6 shows the behaviour of monthly evaporation rates for the four temporal slices: historical (1961-2005), near term (2006-2040), midterm (2041-2070), and long term (2071-2099). The two scenarios derived from the Eta-MIROC5 model (M4 and M8) presented similar behaviour, varying only in the magnitude of the evaporation rates. In summary, both scenarios predict a reduction in the evaporative rate in the first two months of the year, stability until the middle of the dry season, and a reduction in the last two months of the year. April was the only month that showed a consistent increase in evaporative rate,

3% and 4% for the scenarios M4 and M8, respectively. The months of November and December show the greatest expected reduction in evaporation, ranging from -3% to -10%, respectively.

      The scenarios derived from the Eta-CanESM2 model (C4 and C8) show a possible increase in evaporation rate throughout the whole year. However, the greatest increment is expected in the rainy season for the long-term future (2071-2099) under the C8 scenario. The largest absolute variations are expected in March and April of long-term future (8 and 10% increase respectively). The C4 scenario shows stability for the near future (2006-2040) in the dry (from -1% in September to

1% in December) and in the wet seasons (-1% in February and 2% in March and April). The C8 scenario also shows the largest absolute variations in March and April, the increase is 34% and 35% respectively, for the long-term future. The differing evaporation patterns between seasons may be attributed to a combination of factors, such as the decrease in relative humidity and the rise in temperature during the rainy season (Qin et al., 2021). Alternatively, or in conjunction, this behaviour could

also be influenced by a reduction in the frequency of rainy days or cloud cover. Moreover, the downscaling done by the regional models for the C4 and C8 scenarios indicate that the wind speed is expected to increase in the rainy season and decrease in the summer season. The decrease in relative humidity occurs systematically throughout the year, but more pronounced in the rainy season.

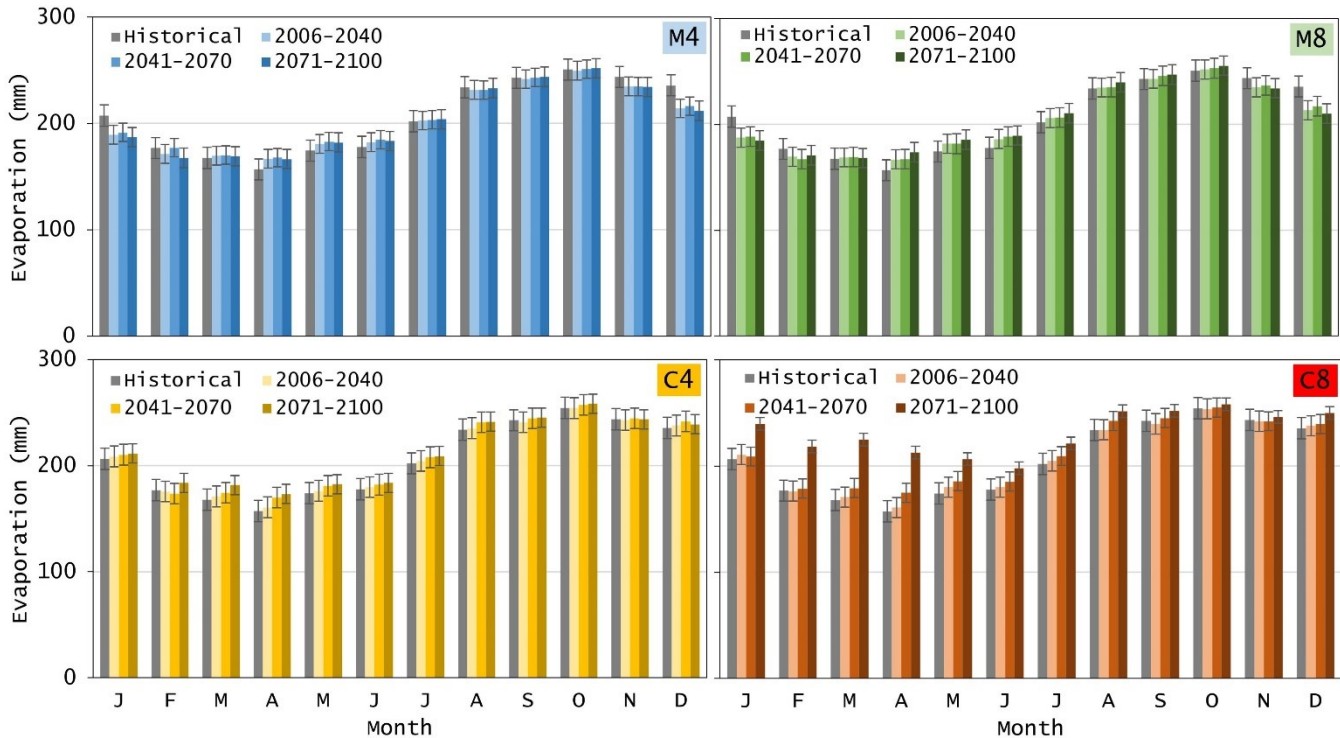

**Figure 6: Scenarios of monthly evaporation rates for the Historical (1961-2005) and RCPs (2006-2099) periods on the Metropolitan Region of Fortaleza. The rates refer to on-water evaporation (Ew). The bars represent the standard error.**

Figure 7 shows an overview of the annual evaporation rates for the modelled historical periods and the four climate change scenarios. A distinct pattern of the models is evident if the 10-year average is considered: the scenarios C8 indicates an upward behaviour, whereas C4, M4 and M8 shows a stabilisation followed by a decrease. From Figure 7 one can depict that there is an increasing behaviour for all scenarios in the beginning of the 2010s. Afterwards, a downward trend is observed for the M4 and M8 scenarios. Except for C8 (which shows a substantial increase), all scenarios tend to maintain the evaporation rate in the long-term future. Over the last decade of simulation (2090s), the M4 scenario presents a more abrupt decrease, while M8 and C4 remain steady.

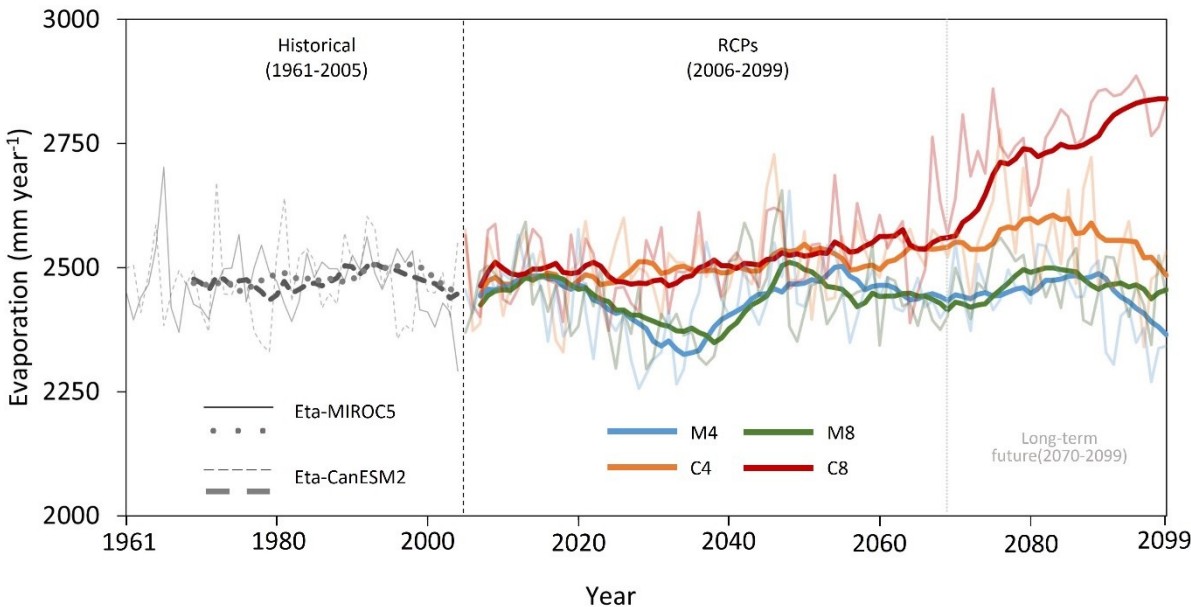

**Figure 7: Simulated annual on-water evaporation for the Metropolitan Region of Fortaleza. The dashed lines represent the simulated Historical (1961-2005) period, and bold lines refer to the 10-year moving averages.**

Table 4 presents the statistical metrics related to the trend analyses of evaporation for all scenarios under the Historical and RCPs experiments. The results indicate no statistically significant variation for the historical simulation of both
models (p-value > 0.05). The M8, C4 and C8 scenarios envisage an increase in the evaporation rate of the reservoirs in the period from 2006 to 2099. However, the Mann-Kendall test detected significant variation simulated by Eta-CanESM2 only: + 0.40 mm yr$^{-1}$ for C4 scenario and + 4.30 mm yr$^{-1}$ for the C8. The M4 scenario is the only one that projects a decrease (-0.01mm yr$^{-1}$) in evaporation, but with no significant trend.

**Table 4.** Mann-Kendall statistics for annual evaporation projected by the regional models for Historical (1961-2005, n = 45), and the RCPs (2006-2099, n = 93) experiments. Bold numbers are statistically significant (p-value < 0.05). The p-values were determined using a two-sided Kendall tau test (Kendall and Gibbons, 1990).

|  | Historical | | Scenarios | | | |
| --- | --- | --- | --- | --- | --- | --- |
|  | **Eta-MIROC5** | **Eta-CanESM2** | **M4** | **M8** | **C4** | **C8** |
| **tau** | 0.008 | 0.010 | -0.003 | 0.105 | 0.192 | 0.582 |
| **p-value** | 0.945 | 0.930 | 0.969 | 0.136 | 0.600 | < 0.001 |
| **Sen's slope (mm yr$^{-1}$)** | 0.07 | 0.04 | -0.01 | 0.42 | 0.40 | **4.30** |

Based on the above findings, we can therefore state that the worst-case scenario in terms of evaporative rate is
320 scenario C8 (+6%) since it increases the evaporative rate in the dry season in the Fortaleza Metropolitan Region. It is worth noting that these rates change when the average annual evaporation, i. e. including both rainy and dry seasons, is considered: -1% under the M4 scenario and +12% under C8.

**4.4 Water availability assessment for the period 2071-2100**

Figure 8 depicts the relation among water yield with their respective annual reliability for the investigated reservoirs in the long-term future. The dashed line refers to the current water availability of the Gavião-Riachão-Pacoti system, whose water yield with 90% annual reliability ($Q_{90}$) is 128 hm³ yr⁻¹. For the scenario C4, the $Q_{90}$ is 119 hm³ yr⁻¹, and for C8 it is 116 hm³ yr⁻¹, which consists in reduction of 6% and 9%, respectively. On the other hand, for the scenario M4, the $Q_{90}$ increases to 129 hm³ yr⁻¹, resulting in 1% of increase in the water availability of the reservoir system. Scenario M8 projects no change in the

future evaporation rate, hence it is not displayed. It is also noteworthy in Figure 7 that, for higher reliability levels (e.g., 99%), there is less variation in water availability between the scenarios, while for smaller reliability levels (e.g., 75%), the differences become more pronounced. That means that the impact of evaporation is greater in regimes of reduced water reliability.

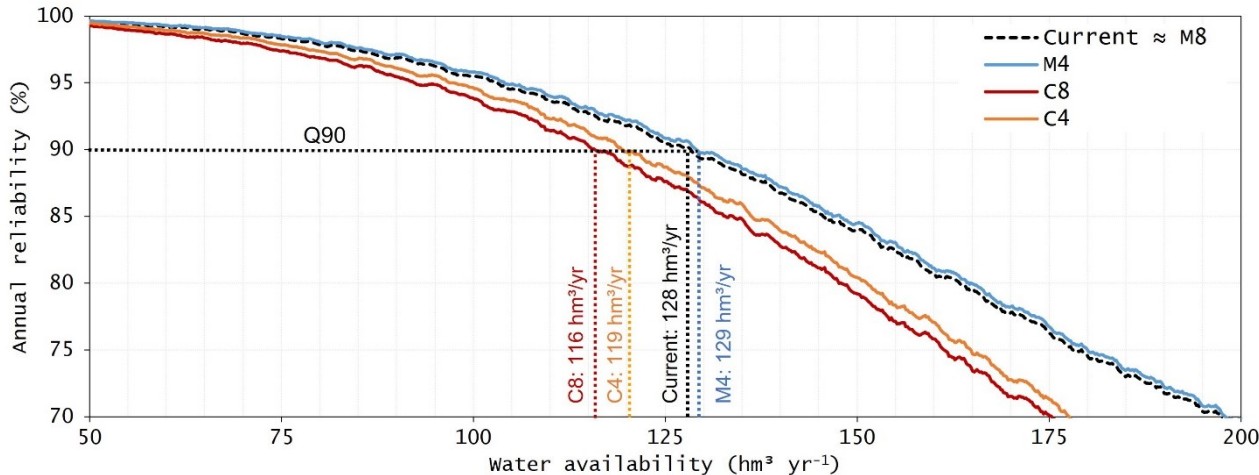

**Figure 8: Water availability as a function of annual reliability level for the Gavião, Riachão and Pacoti reservoirs. Each curve**
**represents a different evaporation rate: current (1961-2005) historical average and four climate change scenarios (M4, M8, C4, and C8) at the end of the 21st Century.**

It is noteworthy in Figure 8 that, although the C8 scenario reports a + 6% change in evaporation rate and the M4 scenario reports -2%, the water availability does not respond linearly to such variations. Table 5 shows the influence of evaporation on

water availability for the case of the Fortaleza Metropolitan Region reservoirs. The results suggest that evaporation significantly influences water availability in the region depending on the scenario. Consequently, the reservoir capacity to supply water with high reliability is reduced. It is relevant to stress that global average elasticity is -0.83, and this value is comparable to siltation impacts observed in reservoirs of the semiarid region (-0.80, as found by de Araújo *et al.*, 2006).

**Table 5.** Elasticity (ε) for the current conditions of water availability in the Metropolitan Region of Fortaleza and for the climatic scenarios in the period 2070-2099. The lower part of the table shows the average elasticity.

| Scenarios | M4 | | M8 | | C4 | | C8 | |
|---|---|---|---|---|---|---|---|---|
| $\Delta E/E^{*}$[1] | $\Delta Q_{90}/Q^{*}_{90}$ | $\varepsilon$ | $\Delta Q_{90}/Q^{*}_{90}$ | $\varepsilon$ | $\Delta Q_{90}/Q^{*}_{90}$ | $\varepsilon$ | $\Delta Q_{90}/Q^{*}_{90}$ | $\varepsilon$ |
| **- 0.15** | 0.099 | -0.657 | 0.092 | -0.615 | 0.052 | -0.346 | 0.023 | -0.153 |
| **- 0.10** | 0.072 | -0.725 | 0.059 | -0.588 | 0.015 | -0.154 | 0.000 | 0.000 |
| **- 0.05** | 0.045 | -0.896 | 0.000 | 0.000 | -0.016 | 0.317 | -0.049 | 0.984 |
| **+ 0.05** | -0.024 | -0.480 | -0.041 | -0.813 | -0.103 | -2.069 | -0.143 | -2.857 |
| **+ 0.10** | -0.058 | -0.579 | -0.085 | -0.847 | -0.153 | -1.532 | -0.185 | -1.852 |
| **+ 0.15** | -0.103 | -0.690 | -0.133 | -0.885 | -0.196 | -1.308 | -0.255 | -1.699 |
| **Average**[2] | -0.014 | -0.674 | -0.040 | -0.627 | -0.091 | -0.949 | -0.126 | -1.085 |

[1] The asterisk refers to the reference values. E* – M4: 977, M8: 997, C4: 1016, C8: 1052 (all values in mm yr[-1]). Q*90 – M4: 136, M8: 134, C4: 119, C8: 116 (all values in hm³ yr[-1])

[2] The global average is -083.

# 5 Discussion

## 5.1 Uncertainties addressed in measurement and modelling

In this research, observed evaporation was assessed in two manners: by the Penman (1948) equation with data from a weather station; and by remote sensing. The station, used to correct the bias in simulated data and to calculate evaporation, is 15 km away from the reservoirs. In most cases, meteorological data are obtained from stations distant from the water body, sometimes tens of kilometres away. Feitosa *et al.* (2021) compared measurements on a reservoir (using a floating raft) with a station located on-land in a distance of 30 km. Their results showed overestimation (71%) of open-water evaporation when using data from the on-land station. Mays (2011) warns that neglecting the impact of reservoir evaporation can result in considerable overestimation of water availability and consequent underestimation of the storage capacity required to support water management decisions. Rodrigues *et al.* (2023) found results supporting this statement: Measurements made directly with sensors in a reservoir showed lower averages than historical measurements or previous studies for the same area (Fortaleza Metropolitan Region). The authors recommend monitoring evaporation based on information obtained from meteorological stations as close as possible to the water body when direct measurements cannot be done.

Since the late 1990s remote sensing tools contribute to the monitoring of water losses (Bastiaanssen et al., 1998; Bastiaanssen, 2000) however the application to lakes and reservoirs is more recent, and this is particularly true for north-eastern Brazil. Except for the investigation of evaporation trends in Brazilian tropical reservoirs by Rodrigues *et al.* (2021a) and the assessment of spatial variability and impact of riparian vegetation by Rodrigues *et al.* (2021b) there are no applications like the one in this research for monitoring open water evaporation with remote sensing. In the present study, we also show the limitations of monitoring evaporation during the rainy season: the use of satellite imagery is impractical since cloudiness impedes visualising the surface. Adding to the limitations in temporal resolution, some reservoirs are rather small and medium spatial resolution images (such as Landsat, 100 x 100 m thermal band and scheduled every 16 days) may show better results

than lower spatial resolution ones (like MODIS, from 500 m thermal bands and scheduled twice a day). This is due to "contamination" of water areas with land area information in large pixels. Further, one should take into consideration limitations due to the absence of field data. This is because AquaSEBS, as well as other models for estimating energy balance and turbulent fluxes (SEBAL, Bastiaanssen et al., 1998; SEBS, Su, 2002), require some information usually acquired on the ground as inputs (e.g. air pressure, temperature, humidity, or wind speed).

Studies report that remote sensing algorithms have a tendency to underestimate evaporation in high-temperature areas (Gokool *et al.*, 2017; Rodrigues *et al.*, 2021a), such as tropical-coastal Northeast Brazil. This feature may have influenced the on-water evaporation, assessed with the help of AquaSEBS. It is shown in Table 3 that the pixel count varies. This is mainly driven by two factors, (i) the presence of clouds and cloud shadows (that were masked out) and (ii) the water level in the reservoirs. The first factor, however, is more strongly related to the final number of pixels used because the reservoirs are maintained full (above 80% of the storage capacity) even in the dry season, which affects the spatial representation less than the gaps produced by the cloudiness. Despite the obvious limitations, we think that the remote sensing approach offers valuable insights - though evidently insufficient to cover the temporal evaporation dynamics, the assessment based on satellite imagery offers an interesting spatial perspective that the few available weather stations cannot provide.

The climate simulations revealed great variations between the regional model outputs and within the historical series of each model. Indeed, climate models mimic the physical mechanisms of the planet, however, data-based representation is not always satisfactorily accurate, particularly when dealing with complex hydrological processes such as lake evaporation. There are external climatic factors such as solar radiation, the Earth's orbit, atmospheric concentrations of greenhouse gases and other atmospheric factors that increase uncertainty. Besides, there are internal factors in the model system that diminish or amplify the effects and generate a high variability (Kundzewicz et al., 2018).

Chou *et al.* (2014) state that, before using climate models as tools to estimate future climate change impacts, the systematic errors of current climate simulations need to be estimated in order to assign some degree of confidence to future climate scenarios. Climate models simulate different future climates which are equally plausible for the same period (in the present case, up to the end of the 21st century). By using bias correction methods, the aim is to bring model simulations closer to real world measurements over a given reference period (baseline). A bias correction method that is valid for the historical validation period should then remain valid for future climate change impact studies (Chen *et al.*, 2020). This assumption has been widely accepted in climate change impact studies (Kundzewicz et al., 2018; Teutschbein and Seibert, 2012; Fiseha et al., 2014). Yet the performance of bias correction can be affected by climate models with different sensitivities to the real system (Chou *et al.*, 2014). Regionally, although driven by a single GCM, various downscaling methods may lead to different future climate scenarios (Adachi and Tomita 2020; Kendon *et al.* 2017; Maraun *et al.* 2015; Tang *et al.* 2016). Downscaling regional climate simulations is another source of uncertainty. While much attention has been paid to the uncertainties of future projections associated with the choices of GCMs (Moges *et al.* 2021), fewer analyses have quantified the downscaling uncertainty (Ahmadalipour *et al.* 2018). Generally, it is recommended to use an ensemble of simulations by multiple GCMs and downscaling techniques for reliable regional climate projections (Dibike *et al.* 2017; Pierce *et al.* 2013).

A notable feature is the uncertainty intrinsic to the climate models themselves. Both simulated similarly the historical period (Eta-MIROC5 with 16.2% difference from the observed data and Eta-CanESM2 with 16.4%). However, when the RCP experiment starts, the four model-derived scenarios behave in different directions. The high variability found in the simulations is attributed to external climate factors such as solar radiation, Earth orbit, atmospheric concentrations of greenhouse gases, and internal factors in the GCMs themselves (Kundzewicz *et al.*, 2018).

CanESM2 (Chylek et al., 2011) and MIROC5 (Watanabe et al., 2010) have some differences in their model configurations, parameterizations, and simulation outputs, namely: i) model physics and dynamics: CanESM2 and MIROC5 use different atmospheric and oceanic dynamical cores, which can lead to differences in the representation of atmospheric and oceanic processes. For example, differences in how convection, cloud formation, and ocean circulation are parameterized can influence simulated climate patterns and variability; ii) forcing scenarios: CanESM2 and MIROC5 may be driven by different historical and future greenhouse gas emissions scenarios and external forcings. Variations in the scenarios used to force the models can lead to differences in simulated climate responses, particularly for future projections of temperature, precipitation, and other climate variables; iii) calibration: Each model undergoes a process of calibration and tuning to ensure that its simulations are consistent with observed climate variability and change. The specific calibration and tuning procedures used for CanESM2 and MIROC5 may differ, leading to differences in model behaviour and performance; iv) data assimilation and initialization: Differences in how observational data are assimilated into the models and how initial conditions are initialized can also lead to divergent simulation outcomes.

These differences between CanESM2 and MIROC5 can contribute to divergent climate projections, particularly at regional scales and for specific study areas. Understanding these distinctions is important for interpreting and contextualizing model results and for assessing the robustness of climate change projections.

Despite the systematic errors inherent in all simulations, the development of regional models for Brazil is essential, as this increases the possibility to better understand the impacts of climate change in various regions, given that it is a country of continental dimensions (8,5 million km²) with climatic, environmental, social, and economic particularities. Our results show that Eta-CanESM2 and Eta-MIROC5 data for Brazil have various biases, which can be originated from the driving GCMs, introduced by the downscaling RCM, and related to uncertainties in observational data. It is expected that such biases, due to the generalised information about the region, are also present in our results, and if the output data are not corrected, any hydrological application will be compromised. Future analyses can be carried out for the study region of this research, such as analysis of the change in the aridity index or the impact of changes in other hydrological processes of relevance for dry regions. The impact on water availability can be reported from two perspectives: first, as a change in reliability for a predefined withdrawal volume and, second, as a change in the available withdrawal volume for a given reliability level (de Araújo *et al.*, 2006). For instance, 128 hm³ yr⁻¹ could be taken from the reservoirs system with 90% reliability under the current climate conditions, but only 116 hm³ yr⁻¹ could be withdrawn with the same reliability under scenario C8 conditions in the period from 2071. This reduction in water availability corresponds to a substantial loss of water resources: under the assumption of per

capita consumption of 150 L day[-1], the difference in water yield would be enough to supply around 200 000 people per year, 6% of the MRF's population.

## 5.2 Evaporation analysis

Our findings show two opposite trends for the same study area: one climate model shows an increase in evaporative rate when compared to the historical period and the other shows a decrease. In fact, there are records around the world of positive and negative trends in evaporation. Liu *et al.* (2004) found that the evaporation of 85 Class A pans in China, between 1955 and 2000, had decreased at an average rate of 29.3 mm per decade. Roderick and Farquhar (2004) observed in regions with large industrial centres in Australia that evaporation reduced by an average of 4.3 mm between 1970 and 2002. Similarly, reductions in evaporation were observed in Canada (Burn and Hesch, 2007), India (Chattopadhyay and Hulme, 1997) and Italy (Moonen *et al.*, 2002). On the contrary, the findings of Zhao *et al.* (2022) show an increasing trend of evaporation at global scale by 0.9% per decade for the period 1985 to 2018. Positive trends in reservoir evaporation were also observed in Benin, West Africa (Hounguè *et al.*, 2019), Austria (Duethmann and Blösch, 2018), Australia (Helfer *et al.*, 2012; Fuentes *et al.*, 2020), centre-west Brazil (Althoff et al., 2019), Czech Republic (Mozny *et al.*, 2020) and in other parts of the world, generally associated with countries or regions with low rates of gas-emissions/industrialization (Wang *et al.*, 2014; Miralles *et al.*, 2014). These different trends of increase and decrease in evaporation in various regions of the planet have been called an "evaporation paradox" (Brutsaert and Parlange, 1998).

While the initial impression of the divergent trends might suggest that model selection alone drives significant water management decisions in a region vulnerable to climate change, it is crucial to acknowledge the complex interplay of factors. Despite the indisputable increase in global temperatures (Solomon et al., 2007; Darshana et al., 2013; Qin et al., 2021), which leads to an anticipated rise in evaporation rates, it is noteworthy that historical simulations from both the Eta-CanESM2 and Eta-MIROC5 models closely align with data recorded by the INMET stations (overestimates of 16.2 % and 16.4 %, respectively). For more details of bias correction of Eta-MIROC5 and EtaCanESM-2 outputs using the Quantile-Mapping method, please refer to Supplementary Material 2 and 3.

Rodrigues *et al.* (2021b) analysed the evaporation trend in Ceará reservoirs for the period 1985 - 2018 using the AquaSEBS model and assessed negative trends (− 0.26 to − 0.08 mm/34 years). According to the authors, such behaviour was attributed to the impact of regional air pollution, analogous to the global dimming effect of reduced evaporation in reservoirs located closer to nearby industrial areas (around 2000, according to IPECE, 2017). This causes an increase in the number of clouds and reduces the influence of solar radiation heating on the evaporation of the water body. However, although Fig. 5 shows a similar pattern to the results of the authors op cit, our results do not have sufficient basis to affirm this. Especially when considering the more continuous modelled data, this trend of a reduction in the evaporative rate is not evident.

One of the main limitations of the original Penman (1948) equation is its simplifying assumptions and empirical coefficients, which may not fully capture the complexities of evaporation processes in tropical climates. For example, the Penman equation assumes constant conditions over a 24-hour period, neglects the effects of diurnal variations in temperature, humidity, and

radiation, and may not adequately account for the specific atmospheric and surface characteristics of tropical regions. However, has undergone adaptations (Valiantzas, 2013) and in this work we strictly follow the steps described by Allen et al. (1998) and McMahon et al. (2013). Studies such as those by Donohue et al. (2010) and Elsawwaf et al. (2010) report that Penman (1948) produces the most realistic estimates of evaporation and is the most comparable to energy balance estimates using the Bowen ratio. In a recent study, Rodrigues et al. (2023) demonstrated to what extent two direct-measurement sensors and two physically-based models (Penman and modified Dalton) accurately estimate the evaporation of a tropical reservoir (same study region of the present paper). The Penman (1948) model, based on data from a floating station, showed good results (r > 0.7) for the 12 h time step or daily evaporation, comparing with the direct measurements.

## 5.3 Water availability

Campos (2010) states that in Brazil, the annual reliability discharge of 90% is commonly used for water resources planning and can be interpreted as the reference water availability of the reservoir. Recio-Villa et al. (2018) also used annual reliability to establish reference water availability, however, they recommend a reliability level of 75% for reservoirs located in a humid tropical climate, which is the case of the region where the reservoirs studied in the present paper are located. Despite this recommendation, we used $Q_{90}$ as a rule for two main reasons: i) the reservoirs of the Metropolitan region of Fortaleza are supplied by a long network of reservoirs located in a semiarid region with water deficit during two thirds of the year and high rainfall uncertainty; ii) the water from these reservoirs is mainly used for industries, agriculture and for the direct supply of about 4 million inhabitants. It should be considered that such a simulated water availability is affected by changes in the evaporation rate. In Ceará, the reservoirs usually suffer a process of reduction in storage capacity mainly due to sediment deposition in the reservoirs (de Araújo et al., 2023) because of erosion; and also due to the high rate of water pollution, mostly caused by eutrophication (Mesquita et al., 2020). Our study highlights another important factor affecting water availability, which is the evaporation rate influenced by climate changes (elasticity concept, as in de Araújo et al., 2006).

As stated by Krol et al. (2003), climate impacts are not merely an effect of changes in water availability but emerge from the confrontation between availability and societal demands, and also the role these demands play in society. This explains why a study should include not only the physical understanding of climate impacts on the water balance, but also the analysis of water use, agricultural economy, and societal impacts. This clearly demonstrates that the study of climate change impacts in developing semiarid regions calls for an integrated approach.

Besides the uncertainty associated with future water availability, there is substantial uncertainty regarding future water demand (Kundzewicz et al., 2018). The findings of de Araújo et al. (2004) indicate that 60 % of the municipalities of the state of Ceará may suffer from long-term water scarcity by 2025. On average, the probability of these municipalities facing a water shortage ranges from 9 % to 20 % annually. Silva et al. (2021) analysed climate change impacts and population growth rates in a basin whose sole reservoir provides water to an urban area of 1 million inhabitants in north-eastern Brazil. The rates of change in population growth for the period from 2015 to 2030 varied between 0.9% and 0.8%. Water recycling and more efficient

technologies can decrease overall water demand. In fact, the results of the investigation by Rodrigues *et al.* (2020) in a tropical reservoir in northeast Brazil indicated that the investment in building a floating photovoltaic power generation system could reduce water losses due to evaporation by approximately $2.6 \times 10^6$ m³ yr$^{-1}$, enough to supply about 50,000 people, whereas the initial investment in the construction would be fully recovered within eight years.

## 5.4 Challenges of an integrated approach to regional water resources and limitations of this research

The present research assessed the impact of evaporation from reservoirs on water availability, although the impact of water quality, silting, and increase in per capita consumption should also be taken into consideration in future investigations. It is necessary, therefore, that water management agencies propose adaptation measures for different scenarios, and this study contributes to decision-making aimed at water security during the dry season. Further investigations in densely-populated areas situated in dry regions may find in these results a reference for studies that take into account other variables which were not 515 addressed in our study.

To achieve an integrated approach for integrating physical climate impacts with societal demands and water use it is first necessary to understand the local context and the specific needs of each region, economic sectors and water resources managers. Then, a comprehensive assessment of the projected impacts of climate change on regional water resources, considering extreme weather events and changes in hydrological regimes. An accurate assessment of the current and future 520 availability of water resources is essential, taking into account surface and groundwater, water quality and its demand. Hydrological models and reliable field data (Rodrigues et al., 2023) to quantify water availability under different climate scenarios are essential.

Identifying social demands and vulnerabilities for water management is crucial. This involves pinpointing key water demands (agricultural, industrial, urban, ecological, recreational) and analysing vulnerabilities such as social inequality and climate 525 resilience. Stakeholder engagement is essential, including local communities, NGOs, private sector, and government. Inclusive participation, interdisciplinary dialogue, and consensus-building are promoted. Continuous monitoring and evaluation of adaptation strategies are necessary to assess their impact on water availability and welfare.

We recommend the work of Sivapalan et al (2012) as a starting point on that matter: "Socio-hydrology: a new science of people and water", and Medeiros and Sivapalan (2020) as complementary reading, whose work evaluates the dynamic nature of 530 human adaptation to droughts since the beginning of the 20th century in the Jaguaribe Basin (89,000 km²) in semiarid Brazil.

## 6 Conclusions

Climate simulations made with the regional models Eta-CanESM2 and Eta-MIROC5 showed different trends in the future evaporation of a reservoir network. Similarly, these trends affected regional water availability of the region in opposing patterns. Four scenarios derived from the RMs were evaluated, and from the results, the following conclusions can be drawn:

- The scenarios derived from the Eta-CanESM2 model (C4 and C8) indicate an increase in dry season evaporative rates: 2% and 6% respectively. When considering the total annual increase, which includes both rainy and dry seasons, the worst-case scenario shows a 12% increase.

- In contrast, scenarios derived from the Eta-MIROC5 model (M4 and M8) display a decrease in dry season evaporative rate. Specifically, the M4 scenario shows a decrease of -2%, while no change is observed in the M8 scenario. However, statistically significant evaporation trends are only observed for the C4 ($+ 0.87$ mm yr$^{-1}$) and C8 ($+ 4.30$ mm yr$^{-1}$) scenarios.

- Regarding the impact of the simulated evaporation on water availability: For a 90% reliability level, the expected range of change in water availability is -7% to +9%. The scenario C8 envisages the highest reduction in annual water flow.

- It is reasonable to state that both patterns of future evaporation in the reservoirs of the Metropolitan Region of Fortaleza are plausible. To reduce uncertainties in modelling future water availability an adaptive management strategy is recommended, in combination with continuous monitoring of climate change and regional development, as it directly affects water demand.

- Although the models used to represent reservoir evaporation are relatively simple and do not account for advective and heat storage effects (which could be enhanced in future investigations), the estimates provided by AquaSEBS are satisfactory. This suggests a promising potential for its application in water resources management.

Given the divergent projections of climate impacts on water resources by different models, adaptive planning should rely on probabilistic approaches using ensembles of projected values rather than single scenarios. Improvements in evaporation measurements are crucial for feeding climate models and remote sensing algorithms, enhancing their reliability. Spatialized field information can further improve regional model simulations by providing basis for bias correction and ground truth. Our findings can supplement water availability estimations, particularly crucial during the dry season (June to December) in north-eastern Brazil.

## Data availability

The historical series observed in Ceará can be obtained from https://bdmep.inmet.gov.br/ in the INMET database. All the CMIP5 RCM outputs are publicly available in the National Institute for Space Research (INPE, Brazil) database, at the INPE Portal "Climate Change in Brazil" and made available on the PROJETA Platform (https://projeta.cptec.inpe.br).

## Author contributions

GP Rodrigues designed the research approach, made the computational analysis and prepared the paper. A Brosinsky elaborated additional research ideas and supervised the work. IS Rodrigues and GL Mamede participated in the design, writing

and revision of the manuscript. JC de Araújo designed the research approach and supervised the work. All authors have reviewed and approved the final version of the manuscript.

## Competing interests

The contact author has declared that neither of the authors has any competing interests.

## Disclaimer

Publisher's note: Copernicus Publications remains neutral with regard to jurisdictional claims in published maps and institutional affiliations.

## Acknowledgements

The authors acknowledge: (1) funding provided by the Brazilian CAPES (Coordination for the Improvement of Higher Education Personnel and the German Academic Research Service (DAAD); (2) FUNCAP (Ceará State Foundation for the
575 Support of Scientific and Technological Development) for the scholarship granted to the first and third authors; (3) COGERH (Water Resources Management Company of Ceará) for the technical support at the Gavião Reservoir; (4) the University of Potsdam for the support during Gláuber Rodrigues's staying at the Institute of Environmental Science and Geography; and (5) Gerd Bürger (University of Potsdam), Saskia Förster (Helmholtz Centre, GFZ) and Birgit Heim (Alfred Wegener Institute, AWI) for the insightful discussion during the preparation of this study. We would like to express our gratitude to the reviewers
(Maarten Krol and an anonymous referee), as well as the editor, Pieter van der Zaag, whose comments significantly enhanced the quality of the paper.

## Financial support

This work was supported by the Coordination for the Improvement of Higher Education Personnel (CAPES) and the German Academic Research Service (DAAD) by means of the public call no. 23/2019; and the Ceará State Research Foundation
(FUNCAP).

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
