# Peer review of "Impact of reservoir evaporation on future water availability in North-Eastern Brazil: A multi-scenario assessment"

_Hydrology and Earth System Sciences, 2023_

## Author Comment (AC1)

**ANSWERS TO REVIEWER #1**

The reviewer's comments are in **black** and our answers in **blue**.

**Reviewer's general comment:** The paper relevantly discusses the potential effects of climate-change-induced changes in reservoir evaporation in a region where water supply critically depends on reservoirs. The paper is clear in its methods, transparent in results and conclusions. Still, some concerns remain, partly on the clarity of the research goal and on some methodological choices.

We are grateful for the reviewer's comments, which are very constructive. Indeed, we agree that some aspects need to be better outlined and that the suggested adjustments will add clarity to our approach and methods.

**Reviewer's comment:** Where the goal of the paper is, to assess the uncertainty in climate change impacts on reservoir evaporation and water availability from reservoirs, it is important to observe that direct climate change effects on evaporation are studied, but climate change effects on hydrology and runoff (reservoir inflow) are not; the character of the study therefore is a sensitivity study on a specific process rather than a more integrated climate change assessment. That does not make the study less relevant, but this difference is important for water availability impacts and should be explicit in the description of the scope or even the title.

**Response:** We agree with the reviewer that the focus of the study is only one hydrological process (open-water evaporation), and possible effects of changes in the evaporative pattern by different future climate scenarios. We highlight that similar analyses are found in the literature, more recently observed in works such as Althoff et al. (2020) in tropical savannah, Andrade et al. (2021) in northeastern Brazil, Viola et al. (2014), Oliveira et al. (2017) in an oceanic temperate climate. The impact of evaporative changes in terms of water availability in a drought-prone overpopulated region. This is relevant because it allows for better adaptation strategies. This is achieved by simulating the evaporation driven by Eta model (Mesinger et al. 2012) outputs from the downscaling of two regional models. Although a similar approach was used in the above-mentioned studies, none of them evaluated water availability impacts influenced by evaporation in a region where there is water pressure. We emphasise the use of the Vyelas model (de Araújo, Güntner, Bronstert, 2006) in estimating water availability, which is a physical model and based on measured data that has been thoroughly monitored for at least 20 years, which endorses the quality of the modelling. We also emphasise the use of the remote sensing algorithm calibrated with field measurements taken at the reservoir.

It was not within the scope of this work to also study the effects on the general hydrology of the region and reservoir inflow, yet we do understand that analyses of the impacts of climate change scenarios on regional hydrology are required to design effective adaptation strategies for specific basins. It is worth mentioning that we are

studying other aspects and factors influencing the evaporation rate in reservoirs (such as the effect of sedimentation and reservoir inflow) in an ongoing research project.

Based on the above, we would like to highlight the insertion of the following text in section 3 Methodology:

*"It is outside the scope of this work to study the effects on the general hydrology of the region, but rather focus on a single hydrological process, which is a major cause of water losses in the region."*

We would like to refer to the text already present in the submitted version, in section 6 Conclusions and outlook:

*"The present research assessed the impact of evaporation from reservoirs on water availability, although the impact of water quality, silting, and increase in per capita consumption should also be taken into consideration in future investigations. It is necessary, therefore, that water management agencies propose adaptation measures for different scenarios, and this study contributes to decision-making aimed at water security during the dry season. Further investigations in densely-populated areas situated in dry regions may find in these results a reference for studies that take into account other variables which were not addressed in our study."*

**Reviewer's comment:** In the research methods, choices for climate scenario data are partly unexplained and partly limitedly connected to the research aims. Results from two GCMs is used; the choice of GCMs is implicit, where an explicit choice was expected, relating to the uncertainty envelope of simulated evaporation trends under climate change: extended the explanation of the choice may resolve this issue. The choice of RCP scenarios (RCP4.5 and RCP 8.5) covers the median to upper range of climate change, where the uncertainty envelope does include more modest changes too. Here an addition of e.g. RCP 2.6 seems to remain consistent with the goal.

**Response:** The **database used** in this research was the one made available by INPE on the PROJETA platform (see https://projeta.cptec.inpe.br), on which almost all climate studies carried out in Brazil are based (see the references in the previous answer). The platform provides parameters downscaled by three global circulation models (CanESM-2, MIROC5 and HadGEM2-ES) forced with the Eta Regional Model, with a spatial resolution of 20km for the study area, and 5km for some regions of Brazil. The products of this database have been successfully validated in several studies for at least 15 years. The **choice of models** was therefore based on previous studies analysing the uncertainty of the models and confirming their suitability for use. In addition, although Eta's spatial resolution is not ideal, it is reasonably sufficient and the best we currently have for the study area. There is a dataset based on an ensemble of 19 bias-corrected CMIP6 climate models projections for the Brazilian territory (CLIMBra - Climate Change Dataset for Brazil, Ballarin et al (2023)). This is an up-to-date and robust dataset; however, it is based on SSP2-4.5 and SSP5-8.5 scenarios. The impacts and vulnerability at regional scale require more detailed climate information. Other global circulation models have lower spatial resolution for the area studied here,

and we favoured using the products scaled by Eta. Furthermore, in accordance with the research objectives, we believe that analysing four possible future **scenarios** is sufficiently effective. However, we agree with the reviewer that the addition of RCP 2.6 would make the results even more robust and perhaps include a third climate model in the assessment. For more detailed information concerning the Eta model we recommend the readers to refer to Chou et al (2014). We expect this to justify to the reviewer the choice of scenarios, models, and RCPs.

The following text will be inserted in Section 3.1 of the paper:

*"The database used in this research is provided by Brazilian National Institute for Space Research (INPE) on the PROJETA platform (see https://projeta.cptec.inpe.br). That is the official source of downscaled climate data for the Brazilian territory with the Eta Regional Model (Mesinger et al., 2014). The latest version of the platform provides parameters downscaled from three global circulation models (CanESM-2, MIROC5 and HadGEM2-ES) forced with the Eta Model with a spatial resolution of 20 km for the study area. The Representative Concentration Pathways available on the platform are 4.5 and 8.5. According to Chou et al. (2014), the Brazilian model has been used operationally at INPE since 1997 for weather forecasts, and since 2002 for seasonal climate forecasts."*

**Reviewer's comment:** One specific assumption in the research methods requires more attention in the form of more discussion or reconsideration. Evaporation over land and over water are linearly related, based on an average of their ratio from (well-)analysed periods. The periods do show a very wide range in that ratio however, and it is likely that the ratio per period analysed depends on drought conditions. Therefore, the relation, if expressed in a stationary ratio, can be expected to be sensitive to climate change. Here at least an extensive discussion is expected; an analysis and possible reconsideration is advised: results may be expected to significantly change.

**Response:** In section 4.2 we present a comparison between on-land (EL) and on-water (Ew) evaporation rates. We made use of images exclusively from the dry season (June to December) due to the following reasons: (i) cloud-free images are easier to obtain in these months; (ii) the water-availability model (Vyelas) only considers evaporation of the dry period; and (iii) this is the period when evaporation is more intense and, thus, more relevant for water management purposes. Average daily evaporation rates generally differ by 27% with on-land evaporation rates being constantly higher than on-water evaporation rates; the correction value KR averages 0.73 (median 0.74). Studies report (Gokool et al., 2017; Rodrigues et al., 2021) that remote sensing algorithms tend to underestimate evaporation in high-temperature areas, such as tropical-coastal Northeast Brazil. This feature may have influenced on-water evaporation which was assessed with the help of AquaSEBS algorithm. We also examined the data in order to detect a possible correlation of KR values with the period of the year when evaporation rates were estimated (for example, higher ratios at the end of the dry season). No correlation was found between the coefficient and the period of assessment, though. Most of the highest KR values (above 0.85) were registered in the first years of monitoring; six of the seven ratios at this threshold are from before 1999. We understand that the revisor suggests that we investigate the influence of drought on the KR ratio. Would you recommend relating KR to a drought

index, such as the SPI (Standardized Precipitation Index)? KR might be sensitive to climate change, but at the moment, our field experiments have provided this value, which will be used hereafter.

**Cited literature, in alphabetic order:**

Althoff et al. (2020): https://doi.org/10.1007/s10584-020-02656-y

Andrade et al. (2021): https://doi.org/10.1002/joc.6751

Ballarin et al. (2023): https://doi.org/10.1038/s41597-023-01956-z

Chou et al. (2014): http://dx.doi.org/10.4236/ajcc.2014.35043

de Araújo, Güntner, Bronstert (2006): https://doi.org/10.1623/hysj.51.1.157

Mesinger et al. (2012): https://doi.org/10.1007/s00703-012-0182-z

Oliveira et al. (2017): https://doi.org/10.1002/joc.5138

Viola et al. (2014): https://doi.org/10.1002/joc.4038

---

## Author Comment (AC2)

**ANSWERS TO REVIEWER #2**

The manuscript addresses the impact of climate change on water resources in Northeast Brazil, a region with significant water scarcity. It integrates a mix of methodologies, including climate modeling using RCP scenarios, remote sensing, and on-site measurements, to analyze evaporation dynamics. This comprehensive approach offers valuable insights for water resource management in the region. However, several concerns need addressing, as outlined below.

We are grateful for the reviewer's comments, which are extremely constructive. Indeed, we agree that many aspects need to be better outlined and that the suggested adjustments will add clarity to our approach and methods.

The reviewer's comments are in **black** and our answers in **blue**.

Major Comments
**1. Uncertainties in Measurement and Modeling**
**1.1 Distance of Meteorological Station (Lines 360-365):**
Considering the cited literature suggesting significant overestimation of open-water evaporation from distant measurements, the use of data from a weather station 20 km away from the reservoirs to correct bias and calculate evaporation is problematic and compromises the overall argument on data accuracy.
**Response:** The meteorological variables did not present discrepancies that significantly alter the estimate of the open-water evaporation rate in stations located in areas with similar and relatively close climatic conditions as in this study (15 km apart). For this response document, we selected 103 measurement days (covering 01/Aug to 01/Dec 2020) and analysed them: The INMET station recorded an average of 6.69 mm/day and stdv of 0.65 mm, and the station mounted on a raft on the reservoir recorded 6.39 mm/day and stdv of 0.75. The difference in accumulated evaporation between the two was 30.0 mm; overall the difference was 4.4%. We believe it is worth highlighting that there are in the State of Ceará (area of 149,000 km²) only 11 climatological stations with long data series (the information is found here, also in English version: https://www.agritempo.gov.br/agritempo/jsp/Estacao/index.jsp?siglaUF=CE&lang=pt_br).

**1.2 Limitations of Remote Sensing Tools (Lines 179-182):**
The use of only 24 satellite scenes over a 24-year period (1994-2018), comprising 18 from Landsat 5 and 6 from Landsat 8, presents a significant limitation for the study's objective of assessing reservoir evaporation trends in the stated region.
**Response:** More scenes were not available and even the available ones are suffering from some cloud coverage. Interestingly, MODIS/MERIS (DAILY acquisition) did not provide very much better coverage. Clearly, cloud cover is an issue in the area. Besides, even with daily acquisition satellites, we faced problems with spatial resolution, even after sharpening the images (see figure). We believe (and would like to know the reviewer's opinion) that obtaining more recent Landsat 8 images (at least two per year, from 2019 to 2023) would add another five years to the series, making it broader and more representative.

[Figure]

Fig 1: On the left, MODIS image with 1km pixels, on the right with 500m resolution.

We have tried also to use the MOD16 product, which provides 500 x 500 m land surface ET datasets for vegetated land areas at 8-day, monthly and annual intervals. The algorithm uses the Penman-Monteith approach to calculate plant and canopy transpiration, as well as soil evaporation (Dias et al., 2021). The pixel values for the two Evapotranspiration layers (ET and PET) are the sum of all eight days within the composite period. We tried to compare AquaSEBS with the PET product,but it was not possible to obtain MODIS scenes on consecutive days due to cloudiness.

Averaging just one scene per year, this frequency (added with consistency issues between the different Landsat products) is arguably insufficient for capturing the complex and dynamic nature of evaporation processes, which are subject to seasonal fluctuations and other meteorological variations. The limitations in using satellite imagery during the rainy season due to cloudiness and their implications on temporal resolution and accuracy of evaporation measurements are acknowledged but not thoroughly analysed.
Response: That is true. However, with regard to the very few available point stations that currently "capture" the temporal variability of evaporation, we think that the remote sensing product adds an interesting perspective on the spatial variability that is clearly evident but invisible to the point stations. Ideally, the approaches should be combined in a way.

The manuscript should address how these limitations impact the overall conclusions. Additionally, employing cloud-penetrating radar or microwave remote sensing (e.g. Synthetic Aperture Radar (SAR)) could have mitigated this limitation.
Response: Interesting approach to think of SAR data. Radar is not affected by clouds and can substitute or replace optical data in many respects but not thermal data that are required for evaporation assessment. We are not aware of a remote sensing algorithm to assess evaporation based on RADAR but would be most interested to learn if we missed a contribution on the topic.
We are aware of two investigations on this topic: one which estimates evaporation of groundwater (Wadge and Archer, 2003), and other investigates soil surface moisture

estimation using ENVISAT ASAR radar data for soil evaporation evaluation (Zribi et al, 2011).

SAR is not typically used to directly measure evaporation from open water surfaces due to its operating frequency and the physical properties of water. SAR primarily detects microwave radiation, which interacts differently with different types of surfaces. Evaporation from open water surfaces involves the process of liquid water transitioning into water vapor due to heat energy from the environment. SAR is not sensitive to this process directly because water in its liquid state has relatively low emissivity and does not scatter microwave radiation significantly.

While SAR itself may not directly measure evaporation rates, it can contribute valuable information to studies investigating for example: water body dynamics (SAR can monitor changes in water extent, water level, and water movement over time. Understanding these dynamics can help estimate water loss through evaporation, especially when combined with other data sources such as meteorological data); and calibration and validation: SAR data can be used in conjunction with other remote sensing data (e.g., optical imagery, thermal infrared imagery) and in situ measurements to calibrate and validate evaporation models and estimates derived from other sensors.

Integrating SAR data with other datasets and models could indeed improve our understanding of this complex process. With regard to limitations and conclusions, there will likely never be a purely EO-based assessment of evaporation in this area of the world. However, we think that EO could support "traditional" evaporation point measurements, which currently occur in Ceará with class A pan measurements, water balance or equations based on measurements from meteorological stations. These methods require different parameterizations. The difficulty in characterizing these parameters makes such approaches complex to use under operational conditions, or in regions with limited ground-truth measurements.

**1.3 The manuscript's approach of comparing point data from a single meteorological station with pixel data from remote sensing poses significant challenges that are not adequately addressed.**

Point data is highly localized and may not represent wider regional conditions, while pixel data offers a broader, albeit less detailed, view. Without proper integration and analysis methodologies, this disparity can lead to misinterpretations or oversights in understanding regional phenomena like evaporation.

The manuscript's lack of discussion on how it reconciles these two data types is a notable omission.

**Response:** We used a floating raft with an on-board meteorological station to compare with the calculations made by the AquaSEBS algorithm. When writing this answer, we decided to compare the KR (0.73) with the real AquaSEBS measurements. We must draw attention to the lower values, which were obtained on cloudy days, which may have interfered with the quality of the image. It can be seen that the values are not so disparate.

Here, we have selected a few acquisitions to compare and present to the reviewer. Perhaps, presenting the data in this way in the finalised and revised document could be more useful and informative to the readers.

Table 1: Comparison between AquaSEBS acquisitions and hypothetical evaporation using KR = 0.73. All values are in mm day[-1]

| Date | On-land | On-water/raft | AquaSEBS | $E_{land}/E_{water}$ | Hypothetical AquaSEBS ($E_{land}$ *KR) |
|---|---|---|---|---|---|
| 22 out 2019 | 7.60 | 6.37 | 4.50 | 0.84 | 5.55 |
| 18 jun 2020 | 7.57 | 6.27 | 4.50 | 0.83 | 5.53 |
| 23 jul 2021 | 7.55 | 7.62 | 7.92 | 1.01 | 5.51 |
| Average | 7.57 | 6.75 | 5.64 | 0.89 | 5.53 |
| Stdev | 0.03 | 0.75 | 1.97 | 0.10 | 0.02 |

We would like to highlight that these data are the best we have for the region, and surely remote sensing tools and efforts like this investigation can provide valuable information for improving evaporation estimates, especially in regions with limited meteorological data coverage. Satellite-based observations can provide information on water surface area, water level fluctuations, and changes in water temperature, aiding in the estimation of evaporation from open water surfaces. Remote sensing data can also be integrated with hydrological models, land surface models, and evapotranspiration models to improve spatially distributed evaporation estimates. Assimilating remote sensing observations into model simulations can enhance the accuracy and spatial resolution of evaporation estimates, especially in regions with heterogeneous landscapes and limited ground-based observations. Remote sensing-derived evaporation estimates can be validated and calibrated using ground-based measurements from meteorological stations. Comparing remote sensing-derived estimates with in-situ measurements allows for the evaluation of model performance and the identification of uncertainties. A less mentioned but important issue is data accessibility and cost-effectiveness: Remote sensing data are increasingly accessible through open-access platforms, making them a cost-effective solution for obtaining spatially distributed evaporation information in data-scarce regions, such as the Brazilian semiarid. Even though our measurements are not ideal, we believe that by leveraging remote sensing tools and integrating satellite observations with traditional meteorological data sources, researchers can enhance spatially distributed evaporation estimates, providing valuable insights for water resource management, climate studies, and environmental monitoring in regions with limited meteorological data coverage.

**2. Climate Model Uncertainties and Bias Correction (Section 5.1, Lines 390-405):**

**2.1 Given the high variability between regional model outputs and the historical series for each model which indicates considerable uncertainties (as described in the manuscript) the manuscript's use of the Linear Scaling Method (LSM) for bias correction of climate model outputs presents notable limitations.**
LSM's simplistic approach assumes stationarity and may inadequately represent extreme weather events and the intricate interactions between various climatic factors. The effectiveness and limitations of the bias correction methods used need a more critical examination.

**Response:** The reviewer's point is valid, LSM is a straightforward and commonly used approach for bias correction in hydrology and climatology studies, however it is often used in studies of hydrological impacts of climate change (to name some Althoff et al., 2020; Oliveira et al., 2017; Fiseha et al., 2014). For the purposes of this study, as well as those mentioned above, the method is sufficiently satisfactory in hydrology. Moreover, its usefulness in data-poor environments is remarkable. In regions with limited observational data or where alternative bias correction methods may be impractical, the LSM can provide reasonable corrections that improve the reliability of climate model outputs or observational analyses (Maraun et al., 2018). While the LSM may not fully capture the complexities of biases in climatic data and may not always yield optimal corrections, its benefits in terms of simplicity, transparency, and versatility make it a valuable tool for bias correction in various climate studies and applications.

In any case, for the final version, we can present as supplementary material a comparison of our results with the more sophisticated Quantile-Mapping method (Teutschbein and Seibert, 2012).

**3. Evaporation Analysis and Water Availability (Sections 5.2 and 5.3, Lines 430-470):**

**3.1 Opposing Trends in Evaporation** - The observation of two opposite trends of evaporation estimates in the study area based on different models is significant. However, the manuscript lacks a detailed analysis of the potential reasons behind these divergent trends, and nor did it present more model ensembles to weigh on a more plausible direction.

**Response:** We agree with the reviewer that an ensemble would be more appropriate for such a divergent result. In fact, in one of the phases of the study, we ran an ensemble of the models and found, albeit less significantly, an upward trend was observed (+3% in the dry season, compared to +6% in the worst-case scenario). The results will be in the revised paper. Indeed, it is common practice in climate research to compare and evaluate multiple models and ensembles to identify commonalities and uncertainties in simulated climate responses. Moreover, we will include the following text in section "5.1 Uncertainties addressed in measurement and modelling":

*"CanESM2 (Chylek et al., 2011) and MIROC5 (Watanabe et al., 2010) have some differences in their model configurations, parameterizations, and simulation outputs, namely: i) Model Physics and Dynamics: CanESM2 and MIROC5 use different atmospheric and oceanic dynamical cores, which can lead to differences in the representation of atmospheric and oceanic processes. For example, differences in how convection, cloud formation, and ocean circulation are parameterized can influence simulated climate patterns and variability; ii) Forcing Scenarios: CanESM2 and MIROC5 may be driven by different historical and future greenhouse gas emissions scenarios and external forcings. Variations in the scenarios used to force the models can lead to differences in simulated climate responses, particularly for future projections of temperature, precipitation, and other climate variables; iii) Model Calibration: Each model undergoes a process of calibration and tuning to ensure that its simulations are consistent with observed climate variability and change. The specific calibration and tuning procedures used for CANESM2 and MIROC5 may differ, leading to differences in model behaviour and performance; iv) Data Assimilation and Initialization: Differences in how observational data are assimilated into the models and how initial conditions are initialized can*

*also lead to divergent simulation outcomes. Variations in data assimilation techniques and initialization procedures can affect the skill and reliability of model simulations.*

*These differences between CanESM2 and MIROC5 can contribute to divergent climate projections, particularly at regional scales and for specific study areas. Understanding these distinctions is important for interpreting and contextualizing model results and for assessing the robustness of climate change projections."*

**3.2 The Penman method may not fully capture the complex evaporation dynamics in a tropical region.** The paper should compare its performance with other methods, like Penman-Monteith, in this specific climatic context, or at least discuss its appropriateness.

**Response:** Indeed, one of the main limitations of the original Penman (1948) equation is its simplifying assumptions and empirical coefficients, which may not fully capture the complexities of evaporation processes in tropical climates. For example, the Penman equation assumes constant conditions over a 24-hour period, neglects the effects of diurnal variations in temperature, humidity, and radiation, and may not adequately account for the specific atmospheric and surface characteristics of tropical regions. However, has undergone adaptations (Valiantzas 2013) and in this work we strictly follow the steps described by Allen et al. (1998) and McMahon et al. (2013). Studies such as those by Donohue et al. (2010) and Elsawwaf et al. (2010) report that Penman (1948) produces the most realistic estimates of evaporation and is the most comparable to energy balance estimates using the Bowen ratio. In a recent study, Rodrigues et al. (2023) demonstrated to what extent two direct-measurement sensors and two physically-based models (Penman and modified Dalton) accurately estimate the evaporation of a tropical reservoir (same study region of the present paper). The Penman (1948) model, based on data from a floating station, showed good results (r > 0.7) for the 12 h time step or daily evaporation, comparing with the direct measurements.

**Minor Comments**

**Future Research Directions (Section 6, Lines 510-515):**
The manuscript highlights the need for an integrated approach but does not provide a clear roadmap for integrating physical climate impacts with societal demands and water use analysis (e.g. RCP-SSP scenarios instead of just RCPs).

**Response:** We believe this is extremely important, but it is outside the scope of this research. Nevertheless, we will insert the following text into the discussion section of the revised paper:

*"To achieve an integrated approach for integrating physical climate impacts with societal demands and water use it is first necessary to understand the local context and the specific needs of each region, economic sectors and water resource managers. Then, a comprehensive assessment of the projected impacts of climate change on regional water resources, considering extreme weather events and changes in hydrological regimes. An accurate assessment of the current and future availability of water resources is essential, taking into account surface and groundwater, water quality and its demand. Hydrological models and reliable field data to quantify water availability under different climate scenarios are essential. Identifying social demands and vulnerabilities is crucial, so the main social demands for water should be identified, including agricultural, industrial, urban, ecological, and recreational use,*

*etc. Analysing social vulnerabilities related to water availability, considering factors such as poverty, social inequality, access to water resources and resilience to climate change. Vulnerable populations, marginalised groups, and areas prone to adverse impacts from water scarcity need to be identified.*
*It is necessary to actively involve local stakeholders, including communities, community leaders, non-governmental organisations, the private sector and government authorities, at all stages of the analysis and planning process. Promote inclusive participation, interdisciplinary dialogue and consensus building around adaptation and water management strategies.*
*Finally, there must be constant monitoring and evaluation systems to follow up on the implementation of adaptation strategies and assess their impacts on water availability and welfare. We recommend the work of Sivapalan et al (2012) as a starting point on that matter: "Socio-hydrology: a new science of people and water"."*

Certain parameters in the Penman equation, such as the psychrometric coefficient and latent heat of vaporization, are assumed constants. It's important to justify these assumptions or discuss their potential impact on the results.
**Response:** We are grateful to the reviewer for the observation and note that this is a textual error. The following text will be inserted in Section 3.2.1 of the paper:

*"$\gamma$ is the psychrometric coefficient; $\rho$ is the density of water (1000 kg m$^{-3}$); $\lambda v$ is the latent heat of vaporization The procedure to obtain $\gamma$ and $\lambda$ is described in Annex 2 of the Food and Agriculture Organization of the United Nations (FAO) 56 protocol (Allen et al. 1998)."*

The calibration of the KR coefficient using AquaSEBS and Penman Equation is mentioned, but details on the validation process, error metrics, or comparison with ground-truth data are not provided.
**Response:** No calibration was done but a direct comparison of the methods using different approaches (EO and Penman applied to a nearby climatological station). Our intention was to use an index that would correlate measurements in different environments. Of course, the mean of 0.73 and the median of 0.74 seemed reasonable (for example, Mamede et al. (2012) indicates that the rate of evaporation in a semi-arid reservoir is approximately 0.70 times the evaporation of a class A pan). However, we know that depending on the size of the historical series, this value can also change. We would like to know if there are any suggestions for improvement from the reviewers on this aspect.

**Cited literature, in alphabetic order:**

Donohue et al. (2010) https://doi.org/10.1016/j.jhydrol.2010.03.020

Elsawwaf et al. (2010) https://doi.org/10.1016/j.jhydrol.2010.10.002

Mamede et al. (2012) www.pnas.org/cgi/doi/10.1073/pnas.1200398109

Marqaun et al. (2018) https://doi.org/10.1038/nclimate3418

Sivapalan et al (2012) https://doi.org/10.1002/hyp.8426

Wadge and Archer (2003) https://doi.org/10.1109/TGRS.2003.813747

Zribi et al (2011) https://doi.org/10.5194/hess-15-345-2011

---

## Author Comment (AC3)

This document-response refers to the following comment from Reviewer #2:

**"Given the high variability between regional model outputs and the historical series for each model which indicates considerable uncertainties (as described in the manuscript) the manuscript's use of the Linear Scaling Method (LSM) for bias correction of climate model outputs presents notable limitations.**
**LSM's simplistic approach assumes stationarity and may inadequately represent extreme weather events and the intricate interactions between various climatic factors. The effectiveness and limitations of the bias correction methods used need a more critical examination."**

The Quantile-mapping adjusts the quantiles of the model data distribution to match those of the observed data distribution, potentially providing a more nuanced correction. The observed differences in evaporation patterns between dry (may to dec) and rainy (jan to apr) seasons could be related to the inherent variability of the climate system. In the rainy season, factors such as increased cloud cover, humidity, and precipitation may influence evaporation rates differently compared to the dry season. Quantile-mapping might better capture these seasonal variations compared to Linear Scaling, resulting in more pronounced differences in evaporation patterns between seasons. The differences could also be related to how well each method handles extreme climatic conditions. Quantile-mapping may be more effective at preserving extreme values or capturing non-linear relationships between variables, potentially leading to different evaporation patterns during periods of climatic extremes.

[Figure]

**Fig 1:** Monthly evaporation after bias correction using two methods. The orange lines refer to the Eta-CanESM-2 model and the green lines refer to the Eta-MIROC5.

The difference (see Tables A and B) is greater in the scenarios from the CanESM-2 model (C4 and C8). The bias correction made after reviewer #2's suggestion (with the more sophisticated Quantile-Mapping (QMP) method), shows an increase of around 1% in the results compared to the previous method (LSM). The scenarios from the Eta-MIROC5 model (M4 and M8) show no more than a 0.4% difference, but still a decrease pattern in the evaporation rate.

A) Annual evaporation after the **Linear Scaling** method

|  | Eta-CanESM2 | | | Eta-MIROC5 | | |
|---|---|---|---|---|---|---|
|  | **Historical** | **C4** | **C8** | **Historical** | **M4** | **M8** |
| **Average** | 2472.4 | 2520.1 | 2597.0 | 2469.9 | 2434.2 | 2444.6 |
| **Change %** |  | +1.9% | +5.0% |  | -1.4% | -1.0% |

B) Annual evaporation after the **Quantile-Mapping** method

|  | Eta-CanESM2 | | | Eta-MIROC5 | | |
|---|---|---|---|---|---|---|
|  | **Historical** | **C4** | **C8** | **Historical** | **M4** | **M8** |
| **Average** | 2493.2 | 2559.2 | 2644.1 | 2495.9 | 2451.5 | 2463.4 |
| **Change %** |  | +2.7% | +6.1% |  | -1.8% | -1.3% |

It is noticeable that the behaviour of evaporation after bias correction with the QMP does not differ much from what was obtained earlier in the investigation: there is a scenario of higher increase in the evaporation rate (C8), two scenarios of reduction (M4 and M8) and a scenario of stabilisation (C4). Although the results do not differ substantially and we have demonstrated that the LSM method is adequate to correct the bias of the data from our study area, we will opt for the QMP method in the final manuscript, given that the impact of these changes on future water availability remains to be assessed, since the effects of the evaporation rate are not linear (as described in the paper in the section covering elasticity). In addition, the impact on evaporation shall be quantitatively estimated using Mann-Kendall trend analysis.

[Figure]

**Fig 2:** Annual evaporation after four scenarios of climate change and bias-correction using the Quantile-mapping method. Bold lines are 10-year moving average.

---

## Editor Decision (ED1)

[revised manuscript text omitted]

---

## Author Response (AR2)

**Author's response to the Editor**

We are deeply grateful to the Editor for his suggestions and corrections. We see them as an effort to improve the article and have accepted all of them, highlighting them **in red** in the text.

In addition, we have inserted all the citations that were not in the list of references.

Lastly, we have changed the title of the manuscript to one that has more connection to our research ("Impact of reservoir evaporation on future water availability in North-Eastern Brazil: A multi-scenario assessment") and moved the text of the last Discussion item to a new item (please see section 5.4).

All the authors have reviewed the final version and agree with the changes.